# SCOPE: Selective Conformal Optimized Pairwise LLM Judging

**Sher Badshah** [1]    **Ali Emami** [2]    **Hassan Sajjad** [1]

## Abstract

Large language models (LLMs) are increasingly used as scalable judges in pairwise evaluation, but they remain prone to miscalibration and biases. We propose SCOPE (Selective Conformal Optimized Pairwise Evaluation), a framework that calibrates an acceptance threshold so that, under exchangeability, the error rate among non-abstained judgments is at most a user-specified level $\alpha$. To supply SCOPE with a bias-neutral uncertainty signal, we introduce Bidirectional Preference Entropy (BPE), which queries the judge under both response positions and converts the order-averaged preference probability into an entropy-based score. Across various pairwise judging benchmarks, BPE outperforms standard confidence proxies in calibration and discrimination, while SCOPE consistently satisfies the target risk bound (empirical FDR $\approx$ 0.097–0.099 at $\alpha = 0.10$) and retains substantial coverage. Compared to vanilla baselines, SCOPE accepts up to $2.4\times$ more judgments under the same risk constraint, demonstrating that BPE enables reliable and high-coverage LLM-based evaluation.

## 1. Introduction

Large language models (LLMs) are increasingly used as judges to scale evaluation in modern AI workflows, from reinforcement learning to automated benchmarking and leaderboards (Zheng et al., 2023; Lambert et al., 2025). By replacing costly human preference labels with model-based pairwise comparisons, LLM-as-a-judge enables rapid evaluation. At the same time, once a model's preferences substitute for human judgments, reliability becomes a first-class requirement: even a small, systematic rate of wrong pairwise decisions can distort rankings and the training sig-

nals derived from them. Concrete failure modes include RLHF reward judges that silently bias the trained policy toward stylistic artefacts such as length or verbosity (Saito et al., 2023; Zheng et al., 2023), leaderboard comparisons that flip when two closely matched models are evaluated by a position-biased judge (Shi et al., 2025), and large-scale annotation pipelines whose distilled preferences inherit the judge's self-preference and familiarity biases downstream (Panickssery et al., 2024). What is missing is a principled way to decide when an LLM judgment should be trusted, together with an explicit, user-specified bound on the error rate among the judgments that are actually accepted.

Selective prediction provides a natural abstraction. Rather than forcing a decision on every instance, the judge abstains when uncertainty is high and returns a judgment only when sufficiently confident (Geifman & El-Yaniv, 2017; Chen et al., 2023). Yet applying selective prediction to pairwise LLM judging exposes two fundamental obstacles. First, thresholding confidence scores offers no finite-sample statistical guarantee that a target accepted-set error will be respected at deployment; thresholds tuned to match validation behavior can exceed the desired risk on new samples. Second, uncertainty proxies are contaminated by nuisance variation (Wang et al., 2025d). Pairwise judges exhibit systematic biases such as position bias (Shi et al., 2025; Wang et al., 2025e), and these effects can produce highly confident but incorrect judgments. As a result, naive confidence thresholding can fail to abstain precisely when it should, violating a user's reliability constraint even when average calibration looks reasonable. Conversely, methods that do provide statistical guarantees often rely on conservative confidence bounds such as Clopper-Pearson (Clopper & Pearson, 1934; Wang et al., 2025c) and fixed sequence testing (Bauer, 1991; Jung et al., 2025), which satisfy the constraint by rejecting a large fraction of queries and thereby sacrificing coverage (Wang et al., 2025c).

This work proposes SCOPE (**S**elective **C**onformal **O**ptimized **P**airwise **E**valuation), a framework for selective pairwise judging with finite-sample statistical guarantees. SCOPE builds on selective conformal prediction and risk control methods (Angelopoulos & Bates, 2023; Angelopoulos et al., 2024; Wang et al., 2025b) to calibrate an acceptance threshold such that the error rate among

[1]Faculty of Computer Science, Dalhousie University, Halifax, NS, Canada [2]Department of Computer Science, Emory University, Atlanta, GA, USA. Correspondence to: Sher Badshah <sh545346@dal.ca>.

*Proceedings of the $43^{rd}$ International Conference on Machine Learning*, Seoul, South Korea. PMLR 306, 2026. Copyright 2026 by the author(s).

accepted judgments is at most a user-specified level $\alpha$ under exchangeability. Unlike heuristic thresholding or naive empirical tuning, SCOPE enforces a finite-sample validity condition on the non-abstained judgments.

Guarantees alone, however, are only useful when the threshold responds to genuine ambiguity in the preference rather than presentation bias. To equip SCOPE with a bias-neutral uncertainty signal, we introduce Bidirectional Preference Entropy (BPE). BPE queries the judge under both orderings of the response pair, aligns the two outputs to the same underlying preference, and then aggregates the resulting preference probabilities. An entropy-based score is then applied to this aggregated probability: uncertainty is highest when the judge is close to evenly split between the two responses, and lowest when one response is strongly preferred. By enforcing permutation invariance with respect to response order, BPE mitigates position effects with only two forward passes per pair, yielding uncertainty estimates that better reflect the underlying difficulty of the preference decision. Empirically, BPE improves calibration and discrimination over standard confidence proxies, and SCOPE leverages these scores to accept more judgments while meeting the target risk level across benchmarks and model scales.

**Contributions.** Our contributions are twofold:

- SCOPE: a conformal-based method for selective LLM-based pairwise evaluation, with a finite-sample guarantee that the error rate among accepted judgments is at most $\alpha$ under exchangeability.

- BPE: a bidirectional, permutation-invariant uncertainty estimator that mitigates position effects and improves uncertainty quality over standard confidence proxies.

## 2. Methodology

We formalize SCOPE for pairwise judging with statistical guarantees of alignment with human preferences.

### 2.1. Problem Formulation

Let $\mathcal{X}$ denote the space of evaluation instances, where each $x \in \mathcal{X}$ consists of a user instruction $q$ and a pair of candidate responses $(r_A, r_B)$. Let $\mathcal{Y} = \{A, B\}$ be the label space, where $y = A$ indicates that $r_A$ is preferred and $y = B$ indicates that $r_B$ is preferred. We assume access to samples from a joint distribution $\mathcal{D}$ over $\mathcal{X} \times \mathcal{Y}$, representing ground-truth human preferences.

**Selective prediction.** An LLM judge defines a distribution $P_\theta(y \mid x)$ over labels $\mathcal{Y}$. Rather than always com-

mitting to a prediction, we adopt selective prediction: the model outputs a judgment only when sufficiently confident (Geifman & El-Yaniv, 2017; Chen et al., 2023).

We define an uncertainty scoring function $s : \mathcal{X} \to \mathbb{R}$, where higher values indicate greater uncertainty. Given a threshold $\lambda$, the selective judge accepts predictions with uncertainty below the threshold:

$$f_\lambda(x) = \begin{cases} \hat{y} & \text{if } s(x) \leq \lambda, \\ \perp & \text{otherwise,} \end{cases}$$

where $\hat{y} : \mathcal{X} \to \mathcal{Y}$ is a deterministic prediction rule (instantiated by BPE in Section 2.2) and $\perp$ denotes abstention.

**False Discovery Rate (FDR) Control.** Our objective is to calibrate $\hat{\lambda}$ such that the error rate among the accepted LLM judgments is statistically bounded. Let $y^*$ denote the ground truth label. We define the selection indicator $S(x, \lambda) = \mathbb{I}(s(x) \leq \lambda)$ and the error indicator $E(x) = \mathbb{I}(\hat{y} \neq y^*)$. Borrowing terminology from multiple-testing, we refer to the selection-conditioned error rate as the false discovery rate (FDR), i.e., the expected fraction of incorrect judgments among those the judge accepts. To control the test-time FDR marginally, we bound the ratio of expected errors to expected selections:

$$\text{FDR}(\lambda) = \frac{\mathbb{E}[S(x, \lambda)E(x)]}{\mathbb{E}[S(x, \lambda)]} = P(\hat{y} \neq y^* \mid s(x) \leq \lambda) \leq \alpha. \tag{1}$$

This formulation ensures that the judge maintains a bounded error rate (e.g., $\leq 5\%$ error for $\alpha = 0.05$) across the distribution of queries. Crucially, this statistical validity holds regardless of the model's intrinsic capability as long as the calibration and test data are exchangeable.

### 2.2. Bidirectional Preference Entropy

To calibrate $\lambda$, the scoring function $s(x)$ must reflect true uncertainty. However, existing uncertainty methods are often miscalibrated for pairwise judging (Jung et al., 2025). For instance, standard predictive probability becomes unreliable when the model systematically favors a particular position (Zheng et al., 2023).

To prevent such systematic biases from contaminating $s(x)$, we propose Bidirectional Preference Entropy (BPE), where we aggregate predictions across both positions. Let $x_{\text{fwd}} = (q, r_A, r_B)$ denote the original position and $x_{\text{rev}} = (q, r_B, r_A)$ the swapped position. We compute the probability that the model prefers $r_A$ under each position:

$$p_{\text{fwd}} = P_\theta(r_A \succ r_B \mid x_{\text{fwd}}) = P_\theta(y = A \mid x_{\text{fwd}}),$$
$$p_{\text{rev}} = P_\theta(r_A \succ r_B \mid x_{\text{rev}}) = P_\theta(y = B \mid x_{\text{rev}}).$$

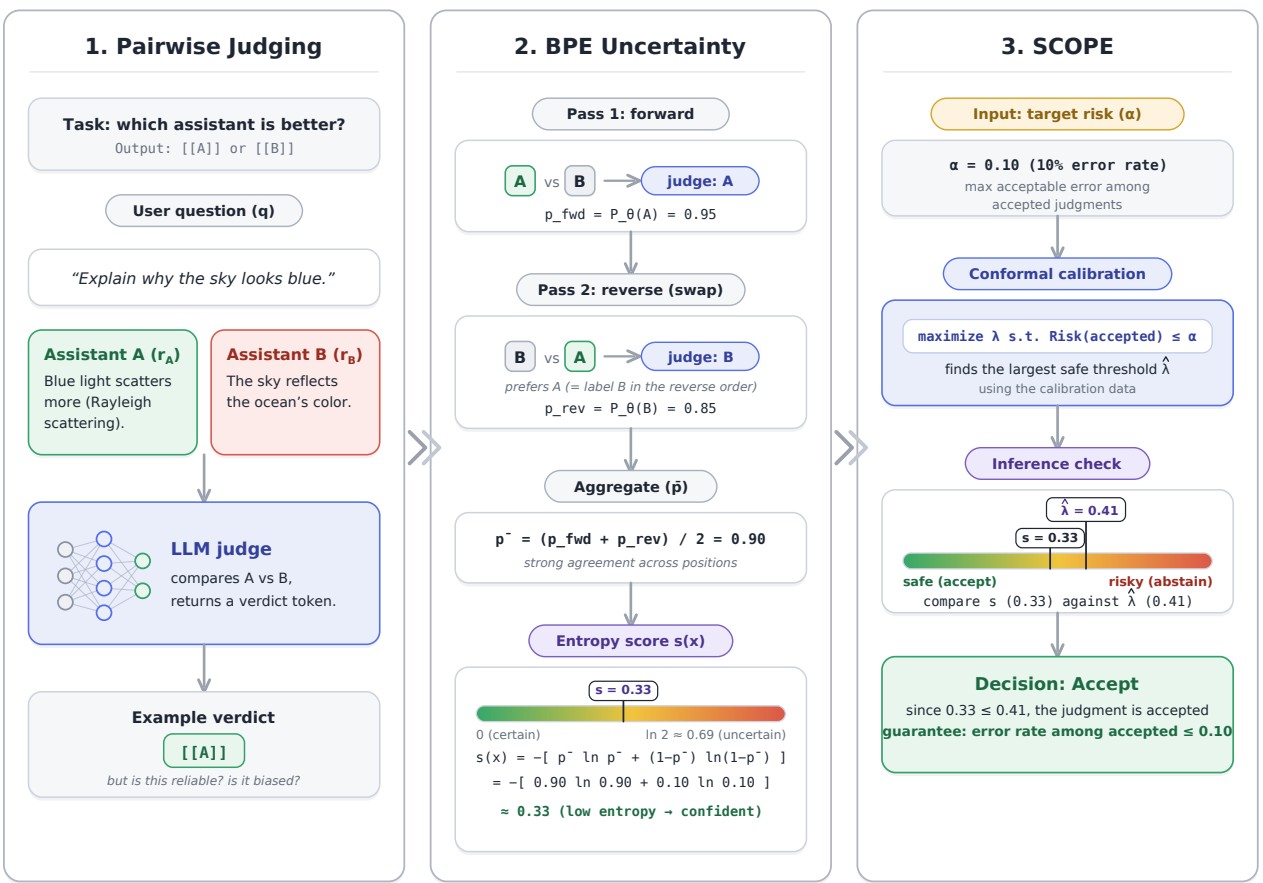

Figure 1. **Overview of the SCOPE framework. (1) Pairwise judging:** for a user query $q$ and two candidate responses $(r_A, r_B)$, the LLM judge returns a single preference token. This raw verdict is sensitive to the order in which the two responses are presented. **(2) Bidirectional Preference Entropy (BPE):** to neutralize potential position bias, the pair is scored in both orders. The forward pass gives $p_{\text{fwd}} = P_\theta(y{=}A \mid x_{\text{fwd}})$ and the reverse pass gives $p_{\text{rev}} = P_\theta(y{=}B \mid x_{\text{rev}})$, since preferring $r_A$ corresponds to label $A$ in the forward order and label $B$ in the reverse order. The two are averaged into a bias-neutral preference $\bar{p} = (p_{\text{fwd}} + p_{\text{rev}})/2$, and its binary entropy $s(x) = -[\bar{p} \ln \bar{p} + (1 - \bar{p}) \ln(1 - \bar{p})]$ (in nats, maximized at $\bar{p}{=}0.5$ with value $\ln 2 \approx 0.69$): small $s(x)$ reflects a confident, order-stable preference, whereas large $s(x)$ reflects disagreement across orders. **(3) SCOPE:** the user specifies a target risk $\alpha$ (e.g., 0.10), the largest acceptable error rate among accepted judgments. Conformal calibration on a labeled set returns the largest threshold $\hat{\lambda}$ certified to keep this error within $\alpha$; at test time the judgment is accepted when $s(x) \leq \hat{\lambda}$ and is abstained otherwise.

Importantly, selecting $r_A$ corresponds to predicting label $A$ in the forward prompt and label $B$ in the reverse prompt. Intuitively, a reliable judge should assign a similar preference probability to $r_A$ regardless of whether it appears first or second; disagreement across permutations is a strong indicator of systematic bias rather than true confidence. We therefore aggregate both by averaging:

$$\bar{p} = \frac{1}{2}\Big(p_{\text{fwd}} + p_{\text{rev}}\Big). \qquad (2)$$

Since the task is binary, the probability of preferring $r_B$ is simply $(1 - \bar{p})$. Because binary entropy is symmetric (i.e., $H(p) = H(1 - p)$), computing uncertainty with respect to $r_A$ suffices without loss of generality.

We derive the final prediction $\hat{y}$ from $\bar{p}$ and define the BPE score as:

$$s(x) = -\left[\bar{p} \log \bar{p} + (1 - \bar{p}) \log(1 - \bar{p})\right]. \qquad (3)$$

The uncertainty score $s(x)$ reaches its maximum when the model provides inconsistent predictions across permutations (see Appendix B.1 for details).

### 2.3. SCOPE Calibration

Given a calibration set $\mathcal{D}_{\text{cal}} = \{(x_i, y_i^*)\}_{i=1}^n$ with ground-truth labels, we seek the threshold $\hat{\lambda}$ that maximizes coverage while ensuring the marginal FDR does not exceed a target $\alpha$.

**Linearization.** Directly controlling the ratio in Eq. 1 is difficult on finite samples: when few predictions are accepted, the denominator $\mathbb{E}[S]$ is small and the ratio becomes unstable. Following Wang et al. (2025b) and Wang et al. (2024b), we reformulate the constraint using a linearized loss for pairwise judging:

$$L(x, \lambda) = S(x, \lambda) \cdot (E(x) - \alpha). \quad (4)$$

The key observation is that $\mathbb{E}[L(x, \lambda)] \leq 0$ implies marginal FDR$(\lambda) \leq \alpha$. This reframes risk control (Angelopoulos et al., 2024) as a budgeting problem where each correct accepted prediction contributes $-\alpha$ to a cumulative sum (building a safety margin), and each incorrect accepted prediction contributes $+(1 - \alpha)$ (depleting the margin).

**Finite-sample calibration.** To guarantee validity on unseen test data, we enforce a finite-sample sufficient condition derived from the theory of linear expectation constraints (Wang et al., 2025b). Specifically, we require the cumulative linearized loss on the calibration set to satisfy:

$$\sum_{i=1}^{n} S(x_i, \lambda) \cdot (E(x_i) - \alpha) \leq -1. \quad (5)$$

Intuitively, the "$-1$" provides a budget that absorbs the worst-case contribution of a single unseen test point: since $L(x, \lambda) = S(x, \lambda)(E(x) - \alpha) \leq 1 - \alpha < 1$, requiring the calibration sum to be at most $-1$ guarantees that even the largest possible test-point loss cannot push the average of $n+1$ linearized losses above zero. This finite-sample correction ensures statistical validity under exchangeability.

**Coverage maximization.** Because newly admitted samples may contribute either $-\alpha$ or $(1 - \alpha)$, the feasibility of Eq. 5 need not be monotone in $\lambda$. We thus select the largest feasible threshold to maximize coverage:

$$\hat{\lambda} = \sup \left\{ \lambda : \sum_{i=1}^{n} S(x_i, \lambda) \cdot (E(x_i) - \alpha) \leq -1 \right\}. \quad (6)$$

If no such $\lambda$ exists (i.e., the set is empty), we set $\hat{\lambda} = -\infty$ and abstain on all instances. This ensures that SCOPE maximizes the number of evaluated instances without violating the risk guarantee.

**BPE preserves exchangeability.** Because BPE is a deterministic mapping $x \mapsto (s(x), \hat{y}(x))$ (two greedy forward passes at $T = 0$), exchangeability of the labeled pairs $(x_i, y_i^*)$ transfers to the induced tuples $(s(x_i), \hat{y}(x_i), y_i^*)$. Consequently, the selection and error indicators $(S(x_i, \lambda), E(x_i))$ entering Eq. 5 are also exchangeable across calibration and test, which is the assumption used in the validity proof.

**Theorem 2.1.** *Let calibration and test samples be exchangeable (Angelopoulos & Bates, 2023). For any $\alpha \in (0, 1)$, the threshold $\hat{\lambda}$ derived in Eq. 6 guarantees that the marginal test-time FDR satisfies:*

$$\frac{\mathbb{E}[E(x_{n+1}) \cdot S(x_{n+1}, \hat{\lambda})]}{\mathbb{E}[S(x_{n+1}, \hat{\lambda})]} \leq \alpha, \quad (7)$$

*where the expectation is taken over the joint randomness of the calibration set and the test sample.*

At test time, for a new instance $x$, we obtain the judge's pairwise evaluation $\hat{y}$ with uncertainty $s(x)$. We accept $\hat{y}$ if and only if $s(x) \leq \hat{\lambda}$; otherwise, we abstain. See Appendix A for the complete proof.

# 3. Experiments

Our experiments address two questions: (1) Does BPE provide better uncertainty estimates than existing methods? (2) Does SCOPE maintain valid risk control while maximizing coverage?

**Datasets.** We evaluate on human-annotated pairwise preferences from three benchmarks: MT-Bench (Zheng et al., 2023), RewardBench (Lambert et al., 2025), and Chatbot Arena (Chiang et al., 2024). The stated benchmarks span diverse evaluation settings: MT-Bench reflects instruction-following and multi-turn assistant quality, RewardBench captures reward-model-style preference signals across heterogeneous sources, and Chatbot Arena represents large-scale crowdsourced user comparisons in open-domain settings.

Following the approach of Jung et al. (2025), we exclude tie outcomes to align with our binary formulation ($\mathcal{Y} = \{A, B\}$). After filtering, we randomly sample $N = 2,000$ non-tied instances to standardize evaluation size and keep the computational cost manageable across our repeated random splits. To probe generalization beyond these three benchmarks, we additionally evaluate on JudgeBench (Tan et al., 2025) and PKU-SafeRLHF (Ji et al., 2025) (see Appendix C).

**Models.** Our judge models span a range of scales: Qwen-2.5-7B-Instruct, Qwen2.5-14B-Instruct, Qwen2.5-32B-Instruct (Yang et al., 2024a), and Llama-3.1-70B-Instruct (Grattafiori et al., 2024).

**Uncertainty quantification.** For BPE, we compute $p_{\text{fwd}}$ and $p_{\text{rev}}$ from the softmax-normalized logits over predictions "A" and "B", requiring two forward passes per instance. For evaluation metrics that expect confidence rather than uncertainty (i.e., ECE, AUROC), we convert BPE into a confidence score via $c(x) = \max(\bar{p}, 1 - \bar{p})$.

**Baselines.** We benchmark BPE against four uncertainty estimation methods for LLM judges, and SCOPE against three selective prediction strategies. Additional comparisons are reported in the Appendix.

**Uncertainty estimation baselines.** The simplest baseline, *Predictive Probability*[1], uses the maximum softmax probability of the zero-shot preference prediction. A complementary, generation-based alternative is *Verbalized Confidence* (Tian et al., 2023), which prompts the judge to directly output a numerical confidence score. The closest relative to BPE is *Swap-and-Aggregate* (Zheng et al., 2023), which queries the judge under both response orderings and uses the discrete agreement rate as confidence. It shares BPE's bidirectional prediction rule but replaces probability-level entropy with a coarse vote-level signal. Finally, *Simulated Annotators* (Jung et al., 2025) estimates confidence via the majority-vote agreement rate among $N = 5$ in-context personas, each conditioned on $K = 5$ few-shot demonstrations sampled from a pool of 50; because this baseline requires multiple generations per instance, we restrict it to Qwen-7B and Qwen-14B judges.

**Selective prediction baselines.** We compare SCOPE against three abstention strategies. *Vanilla* prediction answers all queries without abstention, yielding full coverage but no reliability control. *Heuristic thresholding* is a simple confidence-based rule that accepts predictions whose uncertainty score exceeds $1 - \alpha$, without any calibration guarantee. *Naïve calibration* selects thresholds based on empirical risk measured on held-out data but applies no finite-sample correction and can therefore violate the target risk constraint.

**Evaluation metrics.** We evaluate performance across two dimensions: uncertainty estimation quality and statistical validity of risk control. To assess the performance of uncertainty metrics, we report Accuracy, Expected Calibration Error (ECE) (Naeini et al., 2015), Area Under the ROC Curve (AUROC), and Area Under the Precision-Recall Curve (AUPRC) (see Appendix B).

To validate the selective evaluation framework, we measure the empirical risk (i.e., FDR) on the test set. This value must consistently remain below the target $\alpha$. Finally, we evaluate efficiency via coverage which is defined as the percentage of test queries for which the model returns a prediction rather than abstaining.

**Correctness criteria.** We evaluate the correctness of the LLM judge by comparing its predicted preference against the ground truth human label. For all datasets, a predic-

tion is considered correct if and only if the judge's selected response matches the human-preferred response.

**Protocol.** We utilize a 50/50 split for calibration and test data. To ensure statistical robustness, all reported results are averaged over 1000 independent random splits of the dataset. We evaluate performance across different risk levels $\alpha \in \{0.05, 0.10, 0.15, 0.20, 0.25\}$.

## 4. Results

### 4.1. Uncertainty Estimation Quality

The prerequisite for valid risk control is a scoring function $s(x)$ that effectively ranks judgments by their probability of error. In Table 1, we compare our proposed $s(x)$ against commonly used methods. Across all benchmarks, BPE mostly achieves the highest AUROC and AUPRC, demonstrating capability in distinguishing correct judgments from errors. While we randomize the preference order for all methods to mitigate aggregate position bias, unidirectional metrics such as predictive probability and verbalized confidence remain vulnerable to instance-level overconfidence. BPE also yields lower ECE than predictive probability and verbalized confidence across most configurations, indicating better-calibrated confidence estimates.

We note that the four baselines in Table 1 naturally split into two pairs that share a prediction rule. Predictive probability and Verbalized confidence both use the greedy A/B token from a single zero-shot judge call and differ only in how confidence is computed (max softmax probability versus a separately elicited verbalized score); they therefore report identical accuracy in every row. Swap-and-Aggregate and BPE both aggregate the forward and reverse judge calls and therefore also share accuracy with each other, while differing in how the confidence is computed: S&A uses discrete agreement voting, BPE averages the forward/reverse probabilities and applies binary entropy. Within this bidirectional pair, BPE strictly improves over S&A on ECE, AUROC, and AUPRC in every model–dataset cell, indicating that probability-level aggregation produces a richer uncertainty signal than collapsing each pass to a discrete vote.

Table 2 compares BPE against simulated annotators, a strong baseline that estimates uncertainty through multi-persona agreement. Unlike simulated annotators, which require multiple model generations per instance and are therefore costly to deploy at scale, BPE provides an efficient alternative that achieves stronger uncertainty ranking with only two forward passes. Despite requiring only a single bidirectional probability computation, BPE consistently matches or improves calibration and achieves substantially stronger discrimination across benchmarks. In particular, BPE yields large gains in AUROC and AUPRC

---

[1] We utilize predictive probability for the heuristic and naïve baselines.

*Table 1.* Uncertainty estimation quality. Comparison of BPE against baselines, including the closely related Swap-and-Aggregate (S&A) (Zheng et al., 2023), which shares BPE's bidirectional prediction rule but uses discrete agreement voting as its confidence signal. Bold indicates the best result per model/dataset. BPE achieves superior calibration (ECE ↓) and discrimination (AUPRC) in most settings, and strictly improves over S&A on ECE, AUROC, and AUPRC.

| | MT-Bench | | | | RewardBench | | | | Chatbot Arena | | | |
|---|---|---|---|---|---|---|---|---|---|---|---|---|
| **Method** | Accuracy | ECE↓ | ROC | PRC | Accuracy | ECE↓ | ROC | PRC | Accuracy | ECE↓ | ROC | PRC |
| *Qwen2.5-7B-Instruct* | | | | | | | | | | | | |
| Predictive Probability | 0.731 | 0.239 | 0.658 | 0.824 | 0.757 | 0.210 | 0.727 | 0.891 | 0.751 | 0.218 | 0.709 | 0.865 |
| Verbalized Confidence | 0.731 | 0.146 | 0.497 | 0.737 | 0.757 | 0.128 | 0.496 | 0.765 | 0.751 | **0.129** | 0.477 | 0.747 |
| Swap-and-Aggregate | **0.738** | 0.193 | 0.656 | 0.826 | **0.807** | 0.152 | 0.673 | 0.897 | **0.766** | 0.171 | 0.695 | 0.868 |
| BPE (Ours) | **0.738** | **0.143** | **0.685** | **0.855** | **0.807** | **0.104** | **0.744** | **0.926** | **0.766** | 0.138 | **0.711** | **0.884** |
| *Qwen2.5-14B-Instruct* | | | | | | | | | | | | |
| Predictive Probability | 0.752 | 0.234 | 0.704 | 0.851 | 0.850 | 0.140 | 0.739 | 0.924 | 0.779 | 0.207 | 0.697 | 0.866 |
| Verbalized Confidence | 0.752 | **0.114** | 0.450 | 0.736 | 0.850 | **0.090** | 0.502 | 0.865 | 0.779 | **0.114** | 0.476 | 0.779 |
| Swap-and-Aggregate | **0.766** | 0.204 | 0.679 | 0.851 | **0.874** | 0.113 | 0.704 | 0.929 | **0.790** | 0.177 | 0.680 | 0.867 |
| BPE (Ours) | **0.766** | 0.174 | **0.777** | **0.932** | **0.874** | 0.103 | **0.809** | **0.970** | **0.790** | 0.169 | **0.744** | **0.927** |
| *Qwen2.5-32B-Instruct* | | | | | | | | | | | | |
| Predictive Probability | 0.783 | 0.198 | 0.706 | 0.886 | 0.881 | 0.107 | 0.799 | 0.957 | 0.781 | 0.201 | 0.732 | 0.892 |
| Verbalized Confidence | 0.783 | 0.189 | 0.457 | 0.777 | 0.881 | 0.213 | 0.477 | 0.896 | 0.781 | 0.201 | 0.483 | 0.785 |
| Swap-and-Aggregate | **0.784** | 0.183 | 0.706 | 0.886 | **0.894** | 0.090 | 0.768 | 0.956 | **0.792** | 0.174 | 0.716 | 0.892 |
| BPE (Ours) | **0.784** | **0.172** | **0.729** | **0.908** | **0.894** | **0.071** | **0.822** | **0.975** | **0.792** | **0.151** | **0.742** | **0.919** |
| *Llama-3.1-70B-Instruct* | | | | | | | | | | | | |
| Predictive Probability | 0.755 | 0.185 | 0.741 | 0.887 | 0.866 | 0.092 | 0.752 | 0.947 | 0.802 | 0.138 | 0.735 | 0.913 |
| Verbalized Confidence | 0.755 | **0.096** | 0.484 | 0.761 | 0.866 | **0.059** | 0.485 | 0.873 | 0.802 | **0.073** | 0.472 | 0.809 |
| Swap-and-Aggregate | **0.767** | 0.169 | 0.717 | 0.885 | **0.880** | 0.082 | 0.726 | 0.948 | **0.810** | 0.129 | 0.720 | 0.913 |
| BPE (Ours) | **0.767** | 0.145 | **0.744** | **0.894** | **0.880** | 0.062 | **0.761** | **0.957** | **0.810** | 0.112 | **0.736** | **0.919** |

*Table 2.* Uncertainty estimation quality. BPE outperforms Simulated Annotators (S.A.) in calibration (ECE ↓) and discrimination (AUROC/AUPRC).

| Model / Dataset | Method | ECE↓ | ROC / PRC |
|---|---|---|---|
| **Qwen2.5-7B-Instruct** | | | |
| MT-Bench | S.A. | 0.174 | 0.594 / 0.790 |
| | BPE | **0.143** | **0.685 / 0.855** |
| RewardBench | S.A. | **0.103** | 0.689 / 0.886 |
| | BPE | 0.104 | **0.744 / 0.926** |
| Chatbot Arena | S.A. | 0.158 | 0.646 / 0.804 |
| | BPE | **0.138** | **0.711 / 0.884** |
| **Qwen2.5-14B-Instruct** | | | |
| MT-Bench | S.A. | 0.182 | 0.600 / 0.792 |
| | BPE | **0.174** | **0.777 / 0.932** |
| RewardBench | S.A. | 0.130 | 0.627 / 0.880 |
| | BPE | **0.103** | **0.809 / 0.970** |
| Chatbot Arena | S.A. | 0.176 | 0.608 / 0.788 |
| | BPE | **0.169** | **0.744 / 0.927** |

over simulated annotators on both MT-Bench and Chatbot Arena (e.g., AUROC improving from 0.59 to 0.69 for

Qwen-7B, and from 0.60 to 0.78 for Qwen-14B), indicating a more reliable ranking of error-prone judgments. While simulated annotators can provide competitive calibration in some cases (e.g., RewardBench with Qwen-7B), it remains weaker in separating correct decisions from failures.

### 4.2. SCOPE: Statistical Validity and Coverage

We evaluate whether SCOPE enables selective pairwise evaluation at a user-specified target risk level, i.e., whether the empirical accepted-set error remains below $\alpha$ while retaining as much coverage as possible.

**Baselines violate the risk constraint.** As depicted in Table 3, across given benchmarks, Vanilla prediction (i.e., no abstention) achieves full coverage but substantially exceeds the risk budget (e.g., MT-Bench risk ranges from 0.217–0.269 across models at $\alpha = 0.10$). Similarly, Heuristic thresholding retains high coverage (e.g., 0.809–0.958) yet still violates the constraint in most setting (e.g., MT-Bench risk 0.184–0.251). This gap highlights that raw confidence scores are not reliably calibrated for selective pairwise judging. The Naïve baseline, which tunes a threshold to match the empirical mean on the calibration split, often

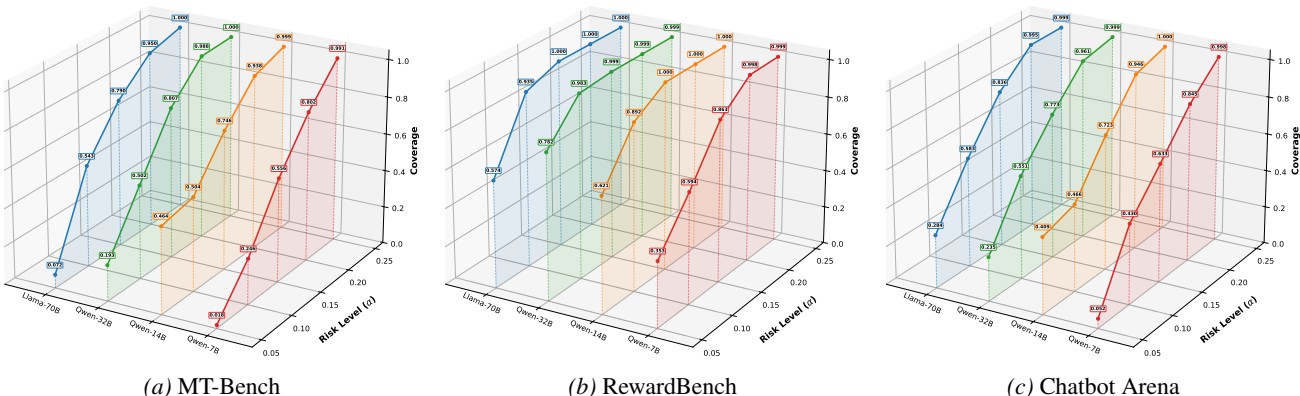

*(a)* MT-Bench        *(b)* RewardBench        *(c)* Chatbot Arena

*Figure 2.* Coverage vs. target risk level $\alpha$ for SCOPE. Coverage increases as the risk budget is relaxed, and larger judges sustain higher coverage at strict tolerances.

*Table 3.* Risk control with SCOPE. Coverage and empirical risk at $\alpha = 0.10$, averaged over 1,000 splits. Values exceeding the risk bound ($> 0.10$) indicate failure. SCOPE consistently satisfies the risk bound while maximizing coverage.

| | Qwen-7B | | Qwen-14B | | Qwen-32B | | Llama-70B | |
| --- | --- | --- | --- | --- | --- | --- | --- | --- |
| | Coverage | Risk | Coverage | Risk | Coverage | Risk | Coverage | Risk |
| *MT-Bench* | | | | | | | | |
| Vanilla | 1.000 | 0.269 | 1.000 | 0.248 | 1.000 | 0.217 | 1.000 | 0.245 |
| Heuristic | 0.907 | 0.251 | 0.953 | 0.231 | 0.933 | 0.200 | 0.809 | 0.184 |
| Naïve | 0.102 | 0.116 | 0.566 | 0.124 | 0.426 | 0.103 | 0.396 | 0.102 |
| SCOPE | **0.246** | **0.097** | **0.504** | **0.098** | **0.502** | **0.099** | **0.543** | **0.098** |
| *RewardBench* | | | | | | | | |
| Vanilla | 1.000 | 0.243 | 1.000 | 0.150 | 1.000 | 0.120 | 1.000 | 0.134 |
| Heuristic | 0.894 | 0.211 | 0.968 | 0.140 | 0.964 | 0.103 | 0.845 | 0.094 |
| Naïve | 0.430 | 0.101 | 0.827 | 0.101 | 0.957 | 0.101 | 0.872 | 0.101 |
| SCOPE | **0.594** | **0.099** | **0.892** | **0.099** | **0.983** | **0.098** | **0.935** | **0.099** |
| *Chatbot Arena* | | | | | | | | |
| Vanilla | 1.000 | 0.249 | 1.000 | 0.221 | 1.000 | 0.219 | 1.000 | 0.198 |
| Heuristic | 0.903 | 0.224 | 0.958 | 0.204 | 0.937 | 0.201 | 0.815 | 0.145 |
| Naïve | 0.297 | 0.103 | 0.530 | 0.114 | 0.532 | 0.102 | 0.565 | 0.101 |
| SCOPE | **0.430** | **0.099** | **0.466** | **0.097** | **0.551** | **0.099** | **0.583** | **0.099** |

operates near the boundary and can fail under finite samples: it exceeds $\alpha$ on MT-Bench for Qwen-7B (0.116) and Qwen-14B (0.124), and on Chatbot Arena for Qwen-14B (0.114). Even when Naïve remains below the bound in some places, it frequently does so by affecting the coverage.

**SCOPE maintains valid risk control while preserving coverage.** Figure 2 shows that SCOPE reliably tracks the target line across $\alpha \in \{0.05, 0.10, 0.15, 0.20, 0.25\}$, keeping empirical risk below $\alpha$ for every dataset and model. At $\alpha = 0.10$ in Table 3, SCOPE achieves risks tightly concentrated around the target (typically 0.097–0.099) while delivering substantially higher coverage than Naïve on challenging benchmarks. For MT-Bench, SCOPE more than doubles coverage for Qwen-7B (0.246 vs. 0.102) and im-

proves coverage for the larger judges as well, all while staying below the 0.10 bound. Overall, SCOPE converts uncertainty estimates into selective judgments that meet the desired risk level, and it does so with higher coverage than empirical thresholding.

**Coverage scales smoothly with the risk budget and model strength.** The coverage-risk curves further illustrate how SCOPE trades off utility against strictness. As the budget relaxes, coverage increases rapidly and approaches full evaluation for stronger judges. For MT-Bench, SCOPE increases coverage from $(0.018, 0.246, 0.556, 0.802, 0.991)$ for Qwen-7B to $(0.072, 0.543, 0.790, 0.950, 1.000)$ for Llama-70B as $\alpha$ ranges from 0.05 to 0.25 (Figure 2a). On Chatbot Arena, the same trend holds, with Qwen-7B moving from 0.052

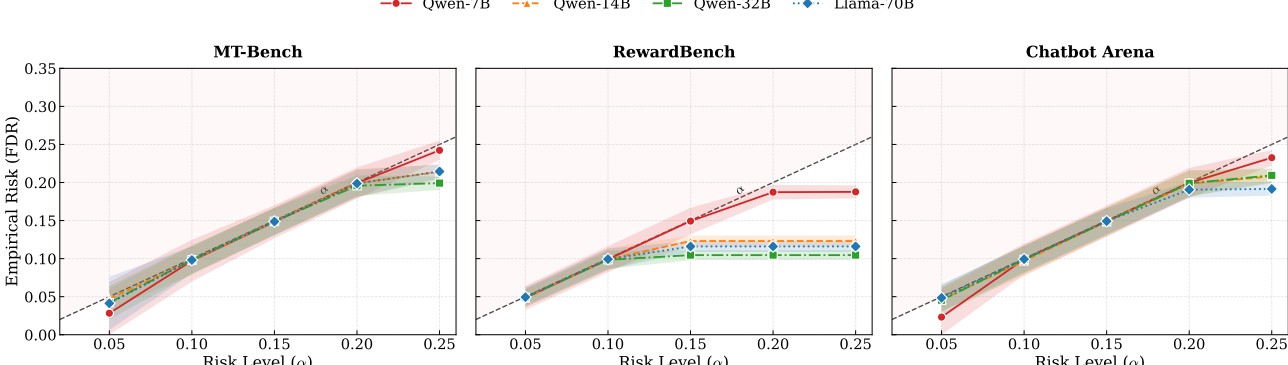

*Figure 3.* Statistical validity of SCOPE across benchmarks. We report the empirical risk (FDR) against the user-specified target risk level $\alpha$. The dashed diagonal line ($y = x$) indicates the theoretical safety limit; curves remaining below this boundary demonstrate valid risk control. Solid lines represent the mean risk over 1000 trials, while shaded regions denote the standard deviation ($\pm 1\sigma$). SCOPE consistently satisfies the risk constraint across judges and tasks.

coverage at $\alpha = 0.05$ to 0.998 at $\alpha = 0.25$, and Llama-70B achieving high coverage even at strict levels (Figure 2c). This monotone behavior at the system level is notable given that feasibility of Eq. 5 need not be monotone in $\lambda$; empirically, the largest-feasible-threshold rule in Eq. 6 yields stable and interpretable tradeoffs.

**Stability across random splits.** Figure 3 reveals how sensitive selective evaluation is to the particular calibration/test split. The shaded bands ($\pm 1\sigma$) are consistently widest for the weakest judge (Qwen-7B), indicating higher variance in accepted-set error across random splits, whereas stronger judges such as Qwen-32B show much tighter bands and more stable behavior. This split-to-split variability is most pronounced at larger $\alpha$, where higher coverage admits more borderline instances and the resulting error rate depends more on which examples fall into the calibration set. Importantly, even in these higher-variance regimes, the mean risk curves remain stable and track the target closely, suggesting that SCOPE is not maintaining validity by being overly conservative; instead, it yields a predictable risk profile even when calibration noise is non-negligible. In other words, SCOPE closely tracks the target risk level $\alpha$, fully utilizing the user-specified risk budget to maximize coverage rather than abstaining too conservatively.

## 5. Related Work

**LLM-as-a-judge.** The high cost of human annotation has driven the adoption of LLMs as scalable surrogates for evaluation (Zheng et al., 2023; Chiang et al., 2023). This paradigm is now widespread in both benchmark-style leaderboards (e.g., MT-Bench) and live preference platforms (e.g., Chatbot Arena) (Chiang et al., 2024). Beyond pairwise win-rates, recent work also uses LLM judges

as general-purpose reference-free metrics for generation quality, often decomposing evaluation into criteria with chain-of-thought or scoring templates (e.g., G-Eval) (Liu et al., 2023). However, a growing body of evidence shows that LLM judges can be systematically unreliable. Documented failure modes include position bias (Wang et al., 2024a; Shi et al., 2025), verbosity/length bias (Zheng et al., 2023; Saito et al., 2023), and self-preference or familiarity biases that favor low-perplexity outputs or the judge's own stylistic priors (Zheng et al., 2023; Panickssery et al., 2024). These issues can distort model rankings and incentivize "judge gaming," especially when benchmarks are optimized against a fixed judge. While mitigation techniques such as swapping positions (Zheng et al., 2023), chain-of-thought prompting (Wei et al., 2022), debate-style judging (Chan et al., 2024), and ensembling or multi-judge aggregation (Badshah et al., 2025; Badshah & Sajjad, 2025) improve empirical agreement, they remain heuristic and do not provide formal reliability guarantees. This naturally shifts attention to uncertainty estimation: an LLM judge should assess when its prediction is likely to be correct and abstain on instances where uncertainty is high.

**Uncertainty estimation for LLM judges.** Several recent works study how to quantify uncertainty in LLM-based evaluations. Xie et al. (2025) conduct a large empirical study of uncertainty in model-based LLM evaluation including pairwise comparison using token probabilities as a proxy for an evaluator's internal confidence and showing that evaluation confidence varies across model families/sizes and is sensitive to distribution shift. Complementarily, Yang et al. (2024b) study verbalized confidence scores, analyzing when self-reported confidence can be reliable and how prompt formulations strongly affect calibration. Most closely related to our baselines, Jung et al. (2025) introduce simulated annotators, where multiple pairwise an-

notations are sampled and agreement is used as a confidence proxy. Another closely related line of work mitigates position bias by re-prompting the judge under swapped or repeated conditions and aggregating decisions, including swap-agreement protocols (Zheng et al., 2023; Wang et al., 2025a; Tripathi et al., 2025) and majority-vote self-consistency (Wang et al., 2023); BPE differs in that it produces a continuous probability-level uncertainty score from just two deterministic passes, rather than a discrete vote over many samples. These lines of work motivate the baseline uncertainty signals used in our experiments. However, without a formal reliability guarantees, these confidence metrics remain heuristic proxies: improvements in calibration or discrimination do not translate into finite-sample, distribution-free guarantees that the error rate among accepted judgments is controlled at a target risk level.

**Conformal prediction and risk control.** Motivated by the above reliability gaps and the lack of formal guarantees, a natural direction is to replace heuristic "confidence" with finite-sample valid uncertainty quantification. Conformal prediction provides a distribution-free framework that converts any scoring function into statistically valid sets via calibration (Vovk et al., 2005; Angelopoulos & Bates, 2023). While classical conformal methods target marginal coverage, recent advances generalize to risk control: instead of guaranteeing set coverage, they guarantee that an error rate on the accepted set is below a user-specified level with high probability (Bates et al., 2021; Angelopoulos et al., 2024). This has enabled selective prediction with rigorous guarantees, where a model abstain to ensure reliability on the non-abstained instances, and more broadly supports controlling discovery-style errors such as false discoveries among accepted decisions (Bates et al., 2021). In the LLM setting, selective prediction have been adapted to reliability in QA and generation, including calibrated abstention and hallucination detection (Gui et al., 2024; Niu et al., 2024; Wang et al., 2025c).

Our work builds on this line by adapting conformal risk control (Angelopoulos et al., 2024) and selective prediction methods such as linear expectation theory (Wang et al., 2025b) to pairwise LLM evaluation, where the objective is not coverage over labels but guaranteeing that the error rate among accepted judgments remains below a target risk level. Prior work typically relies on unidirectional confidence proxies (Wang et al., 2025c), such as maximum softmax probability or sample consistency which can be systematically misaligned with true judging errors due to positional and preference biases (Wang et al., 2025d). We complement conformal risk control with a bias-neutral uncertainty estimator that aggregates preferences under both response positions. This combination enables SCOPE to move beyond heuristic abstention and provides a principled framework for reliable pairwise evaluation, where statisti-cal guarantees are paired with an uncertainty signal tailored to the known failure modes of LLM-based judges.

## 6. Limitations

While SCOPE provides a statistically grounded framework for reliable pairwise LLM judging, several limitations remain. First, our guarantees rely on the standard exchangeability assumption between the calibration set and future evaluation queries. In practice, distribution shifts across benchmarks, prompt variations, or strategic model behaviors may weaken the validity of selective guarantees (Tibshirani et al., 2020; Joshi et al., 2025). Second, BPE requires bidirectional evaluation of each comparison, incurring approximately two forward passes per instance and thus modest computational overhead relative to single-shot confidence heuristics. Moreover, BPE is a white-box uncertainty measure that relies on access to model probabilities or logits, and may not be directly applicable in fully black-box or API-only evaluator settings without approximation. Finally, our formulation focuses on pairwise judging; extending selective guarantees to point-wise evaluation is an important direction for future work.

## 7. Conclusion

As LLMs increasingly serve as judges in many applications, ensuring their reliability is paramount. In this work, we presented SCOPE, a framework that provides rigorous statistical guarantees for LLM-based pairwise evaluation by neutralizing judges' preference bias via our proposed BPE uncertainty estimator and enabling selective evaluation at a user-specified target risk level. Across multiple benchmarks and model scales, BPE improves calibration and discrimination, and SCOPE uses these signals to accept more judgments while meeting the desired risk level, suggesting that combining bias-neutral uncertainty estimation with conformal risk control provides a promising foundation for trustworthy automated evaluation at scale. Crucially, this calibration deploys as a drop-in layer on top of any existing judge without retraining, exposing a single user-tunable knob $\alpha$ that trades coverage against the certified error rate. Looking forward, extending selective guarantees beyond binary pairwise settings to richer paradigms such as multi-response ranking, rubric-based scoring, or interactive critique would broaden applicability, while adapting BPE to fully black-box evaluator settings remains an important challenge for real-world deployment. From RLHF reward modeling to leaderboard rankings, certified abstention provides the accountability layer that purely heuristic judges cannot. Ultimately, we view SCOPE as a step toward principled and trustworthy evaluator systems that can support the next generation of scalable and accountable model assessment.

## Impact Statement

This work advances the reliability of automated model evaluation by introducing a statistically grounded framework for LLM-as-a-judge. To our knowledge, SCOPE is the first framework to provide finite-sample false-discovery-rate control for single-model pairwise LLM judging, and it operates as a drop-in calibration layer on top of any existing judge including specialized, fine-tuned critics without retraining or access to model internals beyond preference-token probabilities. By shifting pairwise judging from heuristic confidence scores to formal risk control, SCOPE enables researchers and practitioners to deploy scalable, low-cost evaluators without sacrificing trustworthiness, and turns LLM judging from an opaque black box into a calibrated system with known, user-tunable reliability properties. Furthermore, the proposed BPE metric actively mitigates position bias, promoting fairer comparisons between models. As LLMs increasingly serve as supervisors for alignment and reinforcement learning, ensuring their judgments come with finite-sample guarantees is a critical step toward accountable and transparent AI development.

## Acknowledgements

We acknowledge the support and funding of CIFAR, the Natural Sciences and Engineering Research Council of Canada (NSERC), the Canada Foundation for Innovation (CFI), and Research Nova Scotia. Advanced computing resources are provided by ACENET, the regional partner in Atlantic Canada, and the Digital Research Alliance of Canada.

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

# A. Proofs of Validity

## A.1. Proof of Theorem 2.1

**Theorem 2.1 (restated).** *Let calibration and test samples be exchangeable. For any $\alpha \in (0,1)$, the threshold $\hat{\lambda}$ defined in Eq. 6 guarantees that the test-time conditional error among accepted predictions satisfies*

$$\frac{\mathbb{E}\left[E(x_{n+1})\, S(x_{n+1}, \hat{\lambda})\right]}{\mathbb{E}\left[S(x_{n+1}, \hat{\lambda})\right]} \le \alpha, \tag{8}$$

*where the expectation is over the joint randomness of the calibration set and the test sample.*

*Proof.* Let $Z_i = (x_i, y_i^*)$ denote a labeled example. For any threshold $\lambda$, recall the selection indicator $S(x, \lambda) = \mathbb{I}(s(x) \le \lambda)$ and the error indicator $E(x) = \mathbb{I}(\hat{y} \ne y^*)$. Define the joint indicator

$$Z(x, \lambda) \triangleq S(x, \lambda)\, E(x) \in \{0, 1\}, \tag{9}$$

and the *linearized loss* (as in Eq. 5)

$$L(Z, \lambda) \triangleq S(x, \lambda)\, (E(x) - \alpha) \;=\; Z(x, \lambda) - \alpha S(x, \lambda). \tag{10}$$

Whenever $\mathbb{E}[S(x_{n+1}, \hat{\lambda})] > 0$, the test-time conditional error rate (our "FDR") can be written as

$$\Pr\!\left(E(x_{n+1}) = 1 \mid S(x_{n+1}, \hat{\lambda}) = 1\right) \;=\; \frac{\mathbb{E}\left[Z(x_{n+1}, \hat{\lambda})\right]}{\mathbb{E}\left[S(x_{n+1}, \hat{\lambda})\right]}. \tag{11}$$

Thus it suffices to show

$$\mathbb{E}\left[L(Z_{n+1}, \hat{\lambda})\right] = \mathbb{E}\left[Z(x_{n+1}, \hat{\lambda}) - \alpha S(x_{n+1}, \hat{\lambda})\right] \le 0, \tag{12}$$

because then rearranging yields $\mathbb{E}[Z(x_{n+1}, \hat{\lambda})]/\mathbb{E}[S(x_{n+1}, \hat{\lambda})] \le \alpha$, which combined with Eq. 11 proves the claim.

**Key exchangeability step.** Consider the joint sample $(Z_1, \ldots, Z_n, Z_{n+1})$, where $Z_{1:n}$ are the calibration examples and $Z_{n+1}$ is the test example. By assumption, these $n + 1$ examples are exchangeable. The calibrated threshold $\hat{\lambda}$ is a (measurable) symmetric function of the calibration set $Z_{1:n}$ via Eq. 6. This is the standard split-conformal swap argument: since $\hat{\lambda}$ depends on $Z_{1:n}$ only through its (unordered) empirical distribution, exchangeability of $(Z_1, \ldots, Z_{n+1})$ implies that, in expectation, any of the $n+1$ points can play the role of the test point. Consequently, the expectation of the linearized loss on the test point equals the expected average of the linearized loss over all $n+1$ points evaluated at the same (random) threshold $\hat{\lambda}$:

$$\mathbb{E}\left[L(Z_{n+1}, \hat{\lambda})\right] \;=\; \mathbb{E}\left[\frac{1}{n+1}\sum_{i=1}^{n+1} L(Z_i, \hat{\lambda})\right]. \tag{13}$$

This is the same identity used in the single-model LEC proof; see Angelopoulos & Bates (2023) for the foundational split-conformal exchangeability argument and Wang et al. (2025b), App. A.1, Eq. (13) for the analogous step.

**Use the calibration constraint.** By construction of $\hat{\lambda}$ (Eq. 6), the calibration sum satisfies the finite-sample sufficient condition (Eq. 5):

$$\sum_{i=1}^{n} L(Z_i, \hat{\lambda}) \le -1. \tag{14}$$

Substituting Eq. 14 into Eq. 13 yields

$$\mathbb{E}\left[L(Z_{n+1}, \hat{\lambda})\right] \;\le\; \mathbb{E}\left[\frac{-1 + L(Z_{n+1}, \hat{\lambda})}{n+1}\right]. \tag{15}$$

**Bound the linearized loss.** Finally, note that since $S \in \{0,1\}$ and $E \in \{0,1\}$, the quantity

$$L(Z, \lambda) = S(x, \lambda)(E(x) - \alpha) = Z(x, \lambda) - \alpha S(x, \lambda)$$

is always strictly less than $1$. Indeed, the maximum occurs when $S = 1$ and $E = 1$, in which case $L = 1 - \alpha < 1$ (since $\alpha > 0$). Therefore,

$$-1 + L(Z_{n+1}, \hat{\lambda}) \leq -1 + (1 - \alpha) = -\alpha < 0. \tag{16}$$

Substituting into Eq. 15 yields

$$\mathbb{E}[L(Z_{n+1}, \hat{\lambda})] \leq -\frac{\alpha}{n+1} < 0,$$

which completes the argument.

**Degenerate case.** If $\mathbb{E}[S(x_{n+1}, \hat{\lambda})] = 0$, then the judge abstains almost surely under $\hat{\lambda}$, and the risk constraint is trivially satisfied (no accepted predictions). This completes the proof.

# B. Experimental Details

## B.1. Bidirectional Preference Entropy (BPE)

We utilize BPE in two distinct forms depending on the downstream application: as an uncertainty score for risk control, and as a confidence score for evaluation benchmarking.

**1. BPE as uncertainty (for SCOPE calibration).** The core of our risk control framework requires a score $s(x)$ where lower values indicate safety and higher values indicate risk. We define this as the binary entropy (in nats) of the bias-neutralized probability $\bar{p}$:

$$s(x) = \text{Entropy}(\bar{p}) = - \left[ \bar{p} \ln \bar{p} + (1 - \bar{p}) \ln(1 - \bar{p}) \right].$$

For example, if $\bar{p} = 0.95$, then $s(x) = -(0.95 \ln 0.95 + 0.05 \ln 0.05) \approx 0.20$, indicating low uncertainty.

This formulation ensures that when the model is maximally uncertain ($\bar{p} = 0.5$), the score is maximized ($s(x) \approx 0.693$), triggering abstention during the calibration process.

**2. BPE as confidence (for metrics).** Standard evaluation metrics such as AUROC and ECE are designed to evaluate *confidence* scores. To benchmark BPE against baselines like predictive probability, we convert the entropy back into a probability-scale confidence score:

$$c_{\text{BPE}}(x) = \max(\bar{p}, 1 - \bar{p}).$$

This maps the uncertainty range $[0.693, 0.0]$ to the confidence range $[0.5, 1.0]$. A score of $1.0$ indicates absolute certainty (low entropy), while $0.5$ indicates a random guess (max entropy). This transformation preserves the rank-ordering of samples, ensuring that AUROC and AUPRC values remain valid comparisons.

## B.2. Evaluation Metrics

We employ four standard metrics to evaluate the quality of uncertainty estimation:

**Accuracy (Acc).** The raw proportion of instances where the judge's selected response ($\hat{y}$) matches the ground truth preference ($y^*$):

$$\text{Acc} = \frac{1}{N} \sum_{i=1}^{N} \mathbb{I}(\hat{y}_i = y_i^*).$$

**Expected Calibration Error (ECE) (Naeini et al., 2015).** Measures the alignment between the model's confidence and its actual accuracy. We bin samples by their confidence score $c(x)$ into $M = 10$ uniform bins. For each bin $B_m$, we compute the average confidence and average accuracy:

$$\text{ECE} = \sum_{m=1}^{M} \frac{|B_m|}{N} \left| \text{acc}(B_m) - \text{conf}(B_m) \right|.$$

Lower ECE indicates that the model's confidence effectively predicts its probability of being correct.

---

**Prompt Template for Pairwise Evaluation**

**[System]**

Please act as an impartial judge and evaluate the quality of the responses provided by two AI assistants to the user question displayed below. You should choose the assistant that follows the user's instructions and answers the user's question better. Your evaluation should consider factors such as the helpfulness, relevance, accuracy, depth, creativity, and level of detail of their responses. Avoid any position biases and ensure that the order in which the responses were presented does not influence your decision. Do not allow the length of the responses to influence your evaluation. Do not favor certain names of the assistants. Be as objective as possible. Output your final verdict by strictly following this format: "`[[A]]`" if assistant A is better, "`[[B]]`" if assistant B is better.

---

**[User Question]**
`{question}`

**[Assistant A's Answer]**
`{answer_a}`

**[Assistant B's Answer]**
`{answer_b}`

---

*Figure 4.* Pairwise evaluation prompt. The system instruction used for all judge models. Instructions requesting an explanation or reasoning trace were removed to enable direct logit extraction (or greedy decoding) of the preference token.

**Area Under the ROC Curve (AUROC).** Measures the ability of the confidence score to distinguish between correct and incorrect predictions, independent of the decision threshold. An AUROC of 1.0 indicates perfect discrimination, while 0.5 indicates random guessing.

**Area Under the Precision-Recall Curve (AUPRC).** Measures the trade-off between precision (correctness) and recall (coverage) as the confidence threshold varies. This is particularly important for selective prediction, as it directly reflects the system's ability to maintain high accuracy at high coverage levels.

### B.3. Implementation Details

To ensure reproducibility, we detail the prompting formats, logit extraction methods, and baseline configurations used in our experiments.

#### B.3.1. PROMPT TEMPLATES

We adopt the standard pairwise evaluation template widely used in MT-Bench and Chatbot Arena (Zheng et al., 2023). The model is instructed to act as an impartial judge and output only the label of the preferred response.

For BPE, we generate the reverse input $x_{\text{rev}}$ by swapping {response_A} and {response_B} in the prompt.

#### B.3.2. LOGIT EXTRACTION

For all models, we compute the uncertainty deterministically. We decode the first token of the response and extract the raw logits corresponding to the tokenizer IDs for "A" and "B". We apply the softmax function exclusively over these two logits to obtain the normalized preference probabilities. This isolates the preference decision from the rest of the vocabulary space. All inference is conducted at temperature $T = 0.0$ (greedy decoding).

#### B.3.3. BASELINE CONFIGURATIONS

**Verbalized confidence:** As illustrated in the Figure 5, we append the following instruction to the base prompt: *"Provide a score between 0.0 (total guess) and 1.0 (absolute certainty)."* The numerical output is parsed directly as the confidence score.

**Simulated Annotators (Jung et al., 2025):** We implement in-context learning ensembles to estimate uncertainty via agreement. For each query, we run the model $N = 5$ times. Each run is conditioned on a distinct annotator persona defined

---

**Prompt Template for Verbalized Confidence**

**[System]**
Please act as an impartial judge and evaluate the quality of the responses provided by two AI assistants to the user question displayed below. You should choose the assistant that follows the user's instructions and answers the user's question better. Your evaluation should consider factors such as the helpfulness, relevance, accuracy, depth, creativity, and level of detail of their responses. Avoid any position biases and ensure that the order in which the responses were presented does not influence your decision. Do not allow the length of the responses to influence your evaluation. Do not favor certain names of the assistants. Be as objective as possible. Output your final verdict by strictly following this format: "`[[A]]`" if assistant A is better, "`[[B]]`" if assistant B is better. **Provide a score between 0.0 (total guess) and 1.0 (absolute certainty).**

---

**[User Question]**
`{question}`

**[Assistant A's Answer]**
`{answer_a}`

**[Assistant B's Answer]**
`{answer_b}`

---

*Figure 5.* Verbalized confidence prompt. As detailed in Appendix B.3.3, the instruction "Provide a score between 0.0 (total guess) and 1.0 (absolute certainty)" is appended to the standard pairwise evaluation prompt to elicit a numerical confidence estimate.

by $K = 5$ few-shot demonstrations, which are sampled from a larger pool of 50 examples. The final confidence score is calculated as the majority agreement ratio among these $N$ personas (see Figure 6 for example prompt).

### B.3.4. INFRASTRUCTURE

All experiments with open-weights models were conducted using the HuggingFace Transformers library (Wolf et al., 2020) on 2×NVIDIA A100 (80GB) GPUs. To ensure statistical robustness, all risk control metrics (i.e., FDR and coverage) are averaged over $1,000$ random $50/50$ calibration-test splits, seeded for deterministic reproducibility.

## C. Extended Benchmarks

To assess generalization beyond MT-Bench, RewardBench, and Chatbot Arena, we evaluate on two additional benchmarks: JudgeBench (Tan et al., 2025), a curated benchmark of pairwise comparisons designed to stress-test LLM judges across knowledge, reasoning, math, and coding domains; and PKU-SafeRLHF (Ji et al., 2025), a safety-focused preference dataset of harmful versus safe responses. Together these probe two regimes underrepresented in the main evaluation: reasoning-intensive judging (JudgeBench) and safety-aligned preference judging (PKU-SafeRLHF). We retain the same protocol as in the main paper ($N$=2,000 non-tied instances per dataset, 1,000 random $50/50$ splits). For consistency across the two extended benchmarks, we report Qwen2.5-7B and Qwen2.5-14B judges, which is the common set for which both benchmarks were processed.

### C.1. Uncertainty Estimation Quality

Table 4 reports the uncertainty estimation quality of BPE against the four baselines on JudgeBench and PKU-SafeRLHF. As in Table 1, BPE achieves the lowest ECE and the highest AUROC/AUPRC in nearly every cell, showing that the calibration and discrimination advantages of probability-level bidirectional aggregation extend to reasoning-heavy and safety-focused settings. The largest absolute gains appear on PKU-SafeRLHF, where BPE improves AUPRC by up to 4.1 points over the strongest baseline. On JudgeBench, where all uncertainty methods struggle (the task is genuinely harder), BPE still delivers the best calibration and the best ranking under AUPRC.

---

**Prompt Template for Simulated Annotators ($i$-th run)**

**[System]**

Please act as an impartial judge and evaluate the quality of the responses provided by two AI assistants to the user question displayed below. You should choose the assistant that follows the user's instructions and answers the user's question better... *[Standard System Instruction]* ... Output your final verdict by strictly following this format: "`[[A]]`" if assistant A is better, "`[[B]]`" if assistant B is better.

---

**[Few-Shot Demonstrations (Persona $i$)]**

*% $K = 5$ examples sampled randomly from the pool of 50*

**[User Question]**
{example_q1}
**[Assistant A's Answer]**
{example_a1} ...
**[Assistant B's Answer]**
{example_b1} ...
**Verdict:** `[[A]]`

*... (repeated for 5 examples) ...*

**[Target User Question]**
{question}

**[Assistant A's Answer]**
{answer_a}

**[Assistant B's Answer]**
{answer_b}

*Figure 6.* Simulated Annotator Prompt. As implemented in the baselines, uncertainty is estimated by agreement. For each query, the model is run $N = 5$ times. Each run is conditioned on a distinct set of $K = 5$ few-shot demonstrations sampled from a pool of 50 examples. To ensure consistency, reasoning traces are omitted, and the few-shot examples follow the exact formatting of the target query.

*Table 4.* Uncertainty estimation quality on the two additional benchmarks. Bold marks the best entry per model/dataset.

| Method | JudgeBench | | | | PKU-SafeRLHF | | | |
|---|---|---|---|---|---|---|---|---|
| | Acc | ECE↓ | ROC | PRC | Acc | ECE↓ | ROC | PRC |
| *Qwen2.5-7B-Instruct* | | | | | | | | |
| Predictive Probability | **0.596** | 0.335 | 0.559 | 0.662 | 0.672 | 0.268 | **0.671** | 0.803 |
| Verbalized Confidence | **0.596** | 0.236 | 0.464 | 0.570 | 0.672 | 0.149 | 0.547 | 0.704 |
| Swap-and-Aggregate | 0.582 | 0.292 | **0.589** | 0.667 | **0.706** | 0.194 | 0.641 | 0.809 |
| BPE (Ours) | 0.582 | **0.174** | 0.583 | **0.716** | **0.706** | **0.136** | 0.648 | **0.820** |
| *Qwen2.5-14B-Instruct* | | | | | | | | |
| Predictive Probability | 0.600 | 0.367 | 0.575 | 0.663 | 0.674 | 0.261 | 0.688 | 0.809 |
| Verbalized Confidence | 0.600 | 0.210 | 0.430 | 0.565 | 0.674 | **0.084** | 0.582 | 0.721 |
| Swap-and-Aggregate | **0.604** | 0.292 | 0.581 | 0.671 | **0.682** | 0.210 | 0.684 | 0.812 |
| BPE (Ours) | **0.604** | **0.232** | **0.597** | **0.734** | **0.682** | 0.150 | **0.689** | **0.845** |

## C.2. Risk Control and Coverage

Table 5 reports coverage and risk for SCOPE at $\alpha = 0.10$ on the two additional benchmarks, and Figure 7 shows the full validity curve across $\alpha \in \{0.05, \dots, 0.30\}$. SCOPE consistently satisfies the risk bound. Coverage scales with judge

capability as expected: stronger judges produce more confidently-correct decisions, yielding more accepted predictions under the same risk budget. PKU-SafeRLHF, where pairwise contrasts between safe and harmful responses are typically clearer, admits higher coverage than JudgeBench at the same model scale and exhibits much tighter split-to-split variance.

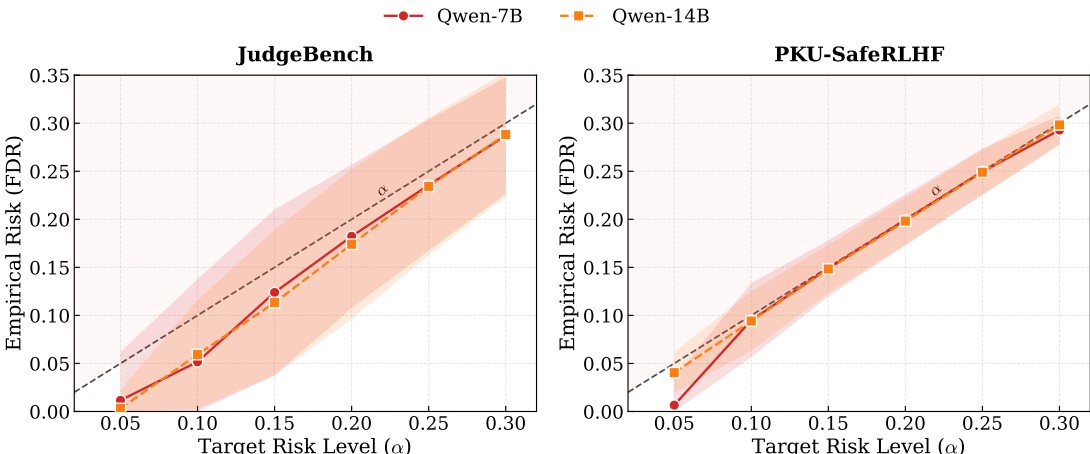

*Figure 7.* Statistical validity of SCOPE on JudgeBench and PKU-SafeRLHF. We report the empirical risk (FDR) against the user-specified target risk level $\alpha$. The dashed diagonal ($y = x$) indicates the theoretical safety limit; curves remaining below it demonstrate valid risk control. Solid lines are the mean risk over 1,000 splits; shaded regions are $\pm 1\sigma$. SCOPE satisfies the risk constraint on both extended benchmarks.

*Table 5.* SCOPE coverage and empirical risk at $\alpha = 0.10$ on JudgeBench and PKU-SafeRLHF (mean over 1,000 random splits; std in parentheses). All empirical risks remain at or below the target.

| | **JudgeBench** | | **PKU-SafeRLHF** | |
| Judge | Coverage | Risk | Coverage | Risk |
|---|---|---|---|---|
| Qwen2.5-7B | 0.023 (0.041) | 0.052 (0.086) | 0.139 (0.049) | 0.095 (0.039) |
| Qwen2.5-14B | 0.114 (0.014) | 0.059 (0.057) | 0.200 (0.036) | 0.094 (0.031) |

# D. Extended Baseline Comparisons

This appendix reports additional comparisons that complement the main results. We emphasize that the metric of interest throughout is the quality of the confidence/uncertainty signal (ECE, AUROC, AUPRC), since SCOPE's calibration consumes a continuous uncertainty score and accepts predictions based on that score; accuracy is reported for transparency but is not the criterion by which selective evaluation methods should be compared.

### D.1. Additional Uncertainty Estimation Baselines

**Self-Consistency.** The method of Wang et al. (2023) aggregate $k$ sampled generations via majority vote, using the vote-agreement ratio as confidence (Lyu et al., 2025). We instantiate $k$=5 at $T$=0.7 and restrict the comparison to Qwen2.5-7B and Qwen2.5-14B judges due to the sampling cost. BPE numbers in Table 6 are reproduced from Table 1 for direct cross-reference.

While SC's majority vote raises accuracy by 7–10 absolute points over greedy BPE, its discrete vote-agreement confidence is unsuitable as input to threshold-based selective evaluation: AUROC values are at or below chance in five of six cells, indicating the signal does not separate correct from incorrect judgments. BPE's continuous entropy is well-ranked (AUROC 0.68–0.81) and better calibrated (lower ECE) at a fraction of the compute (two deterministic passes versus five sampled generations).

*Table 6.* Comparison against Self-Consistency (SC, $k$=5 samples at $T$=0.7). Although SC's majority vote yields higher accuracy, its vote-agreement confidence takes only three discrete values and is uncorrelated or anti-correlated with correctness (AUROC $\leq$ 0.5 in 5 of 6 cells). BPE improves over SC on ECE, AUROC, and AUPRC in every cell. Bold marks the better entry per cell.

| Method | MT-Bench | | | | RewardBench | | | | Chatbot Arena | | | |
|---|---|---|---|---|---|---|---|---|---|---|---|---|
| | Acc | ECE↓ | ROC | PRC | Acc | ECE↓ | ROC | PRC | Acc | ECE↓ | ROC | PRC |
| *Qwen2.5-7B-Instruct* | | | | | | | | | | | | |
| Self-Consistency ($k$=5) | **0.843** | 0.241 | 0.413 | 0.823 | **0.896** | 0.197 | 0.426 | 0.886 | **0.839** | 0.232 | 0.446 | 0.830 |
| BPE (Ours) | 0.738 | **0.143** | **0.685** | **0.855** | 0.807 | **0.104** | **0.744** | **0.926** | 0.766 | **0.138** | **0.711** | **0.884** |
| *Qwen2.5-14B-Instruct* | | | | | | | | | | | | |
| Self-Consistency ($k$=5) | **0.837** | 0.212 | 0.464 | 0.829 | **0.909** | 0.137 | 0.501 | 0.911 | **0.828** | 0.216 | 0.473 | 0.822 |
| BPE (Ours) | 0.766 | **0.174** | **0.777** | **0.932** | 0.874 | **0.103** | **0.809** | **0.970** | 0.790 | **0.169** | **0.744** | **0.927** |

## D.2. Additional Selective Prediction Baselines

**Trust-or-Escalate.** Introduced by Jung et al. (2025), Trust-or-Escalate (ToE) is a selective evaluation framework that uses Clopper–Pearson upper confidence bounds (Clopper & Pearson, 1934) with fixed-sequence testing (Bauer, 1991) to certify the accepted-set error rate. We instantiate ToE with the majority-vote confidence signal from Simulated Annotators ($N$=5, $K$=5), which takes three discrete values $\{0.6, 0.8, 1.0\}$. Because ToE requires $N$ generations per instance and the simulated-annotator pool is dataset-specific, we restrict the comparison to MT-Bench. SCOPE coverage values are reproduced from Table 3.

*Table 7.* Coverage at $\alpha = 0.10$ on MT-Bench for Trust-or-Escalate (ToE) and SCOPE.

| Method | Qwen-7B | Qwen-14B | Qwen-32B | Llama-70B |
|---|---|---|---|---|
| ToE (SimAnnot-Maj) | 0.226 | 0.200 | 0.201 | 0.171 |
| SCOPE (Ours) | **0.246** | **0.504** | **0.502** | **0.543** |

ToE's coverage is capped at 17–23% because its three discrete confidence levels create blocks of identical-confidence items, and the fixed-sequence test can effectively only admit the unanimous-agreement block. SCOPE achieves 1.1–3.2× higher coverage under the same risk constraint by combining BPE's continuous well-spread uncertainty signal with the linear-expectation constraint (Eq. 5), which is less conservative than enforcing a per-prefix UCB.

# E. Drop-in Deployment on a Specialized Critic

To demonstrate that SCOPE is drop-in deployable on existing fine-tuned critics, we evaluate it on Skywork-Critic-Llama-3.1-8B (Shiwen et al., 2024), a publicly released specialized judge fine-tuned from Llama-3.1-8B for pairwise reward modeling. Unlike the general-purpose instruction-tuned models studied in the main paper, Skywork-Critic is purpose-built for the judging task itself, making it a natural stress test for whether BPE and SCOPE transfer to off-the-shelf critics. We use the same three benchmarks as in the main paper (MT-Bench, RewardBench, Chatbot Arena) and the same protocol ($N$=2,000 instances, 1,000 random 50/50 splits).

**Uncertainty estimation quality.** Table 8 reports BPE against the four baselines on Skywork-Critic. Two observations stand out. First, BPE achieves the lowest ECE and the highest AUPRC on all three benchmarks, confirming that probability-level bidirectional aggregation transfers to specialized critics. Second, Verbalized Confidence collapses to AUROC = 0.500 on every dataset. Skywork-Critic, being fine-tuned only for the pairwise decision token, does not produce meaningful self-reported numerical confidence scores. This is a strong demonstration that the choice of uncertainty signal is judge-dependent: a baseline that works on instruction-tuned models can be entirely uninformative on a specialized critic.

**Risk control and coverage.** Table 9 reports SCOPE coverage and empirical risk at $\alpha = 0.10$ on Skywork-Critic across the three benchmarks. All empirical risks remain at or below the target. The most striking result is on RewardBench where SCOPE achieves 99.6% coverage.

*Table 8.* Uncertainty estimation quality on Skywork-Critic-Llama-3.1-8B. Bold marks the best entry per dataset. Verbalized Confidence collapses on this specialized critic; BPE still delivers the best calibration and ranking.

| Method | Acc | ECE↓ | ROC | PRC |
|---|---|---|---|---|
| *MT-Bench* | | | | |
| Predictive Probability | 0.771 | 0.161 | 0.706 | 0.886 |
| Verbalized Confidence | 0.771 | 0.271 | 0.500 | 0.771 |
| Swap-and-Aggregate | **0.776** | 0.152 | 0.696 | 0.886 |
| BPE (Ours) | **0.776** | **0.139** | **0.710** | **0.890** |
| *RewardBench* | | | | |
| Predictive Probability | 0.885 | 0.059 | **0.816** | 0.970 |
| Verbalized Confidence | 0.885 | 0.384 | 0.500 | 0.885 |
| Swap-and-Aggregate | **0.904** | 0.049 | 0.780 | 0.970 |
| BPE (Ours) | **0.904** | **0.039** | 0.795 | **0.972** |
| *Chatbot Arena* | | | | |
| Predictive Probability | 0.759 | 0.175 | **0.711** | 0.881 |
| Verbalized Confidence | 0.759 | 0.259 | 0.500 | 0.759 |
| Swap-and-Aggregate | **0.766** | 0.166 | 0.699 | 0.880 |
| BPE (Ours) | **0.766** | **0.151** | 0.703 | **0.884** |

*Table 9.* SCOPE coverage and empirical risk at $\alpha = 0.10$ on Skywork-Critic-Llama-3.1-8B (mean over 1,000 random splits; std in parentheses). All empirical risks remain at or below the target $\alpha = 0.10$. On RewardBench, SCOPE retains 99.6% coverage.

| Dataset | Coverage | Risk |
|---|---|---|
| MT-Bench | 0.380 (0.064) | 0.098 (0.023) |
| RewardBench | 0.996 (0.008) | 0.094 (0.008) |
| Chatbot Arena | 0.395 (0.050) | 0.098 (0.021) |

*Table 10.* Risk control under label noise on the calibration set ($\alpha = 0.10$, 500 random splits). Each cell shows empirical risk / coverage. SCOPE's empirical risk remains at or below $\alpha$ at every noise level; coverage degrades gracefully as noise increases.

| Judge | Dataset | $\epsilon = 0\%$ | $\epsilon = 5\%$ | $\epsilon = 10\%$ | $\epsilon = 15\%$ | $\epsilon = 20\%$ |
|---|---|---|---|---|---|---|
| Qwen-7B | MT-Bench | 0.098 / 0.248 | 0.046 / 0.059 | 0.020 / 0.015 | 0.007 / 0.004 | 0.002 / 0.001 |
| | RewardBench | 0.098 / 0.590 | 0.055 / 0.380 | 0.010 / 0.092 | 0.000 / 0.003 | 0.000 / 0.000 |
| | Arena | 0.100 / 0.434 | 0.048 / 0.136 | 0.011 / 0.010 | 0.004 / 0.002 | 0.000 / 0.000 |
| Qwen-14B | MT-Bench | 0.098 / 0.506 | 0.053 / 0.467 | 0.003 / 0.137 | 0.000 / 0.000 | 0.000 / 0.000 |
| | RewardBench | 0.099 / 0.891 | 0.055 / 0.668 | 0.003 / 0.195 | 0.000 / 0.000 | 0.000 / 0.000 |
| | Arena | 0.098 / 0.466 | 0.053 / 0.413 | 0.004 / 0.088 | 0.000 / 0.000 | 0.000 / 0.000 |
| Qwen-32B | MT-Bench | 0.100 / 0.506 | 0.049 / 0.230 | 0.006 / 0.054 | 0.000 / 0.001 | 0.000 / 0.000 |
| | RewardBench | 0.098 / 0.982 | 0.055 / 0.816 | 0.005 / 0.154 | 0.000 / 0.000 | 0.000 / 0.000 |
| | Arena | 0.099 / 0.552 | 0.053 / 0.278 | 0.006 / 0.073 | 0.000 / 0.000 | 0.000 / 0.000 |
| Llama-70B | MT-Bench | 0.099 / 0.548 | 0.053 / 0.147 | 0.022 / 0.022 | 0.008 / 0.006 | 0.004 / 0.002 |
| | RewardBench | 0.098 / 0.933 | 0.054 / 0.646 | 0.010 / 0.136 | 0.001 / 0.019 | 0.000 / 0.004 |
| | Arena | 0.100 / 0.584 | 0.056 / 0.331 | 0.020 / 0.064 | 0.005 / 0.009 | 0.002 / 0.002 |

## F. Robustness to Label Noise on the Calibration Set

A natural concern with conformal calibration is that it relies on accurate ground-truth labels on the calibration set; in practice, human preference labels are noisy. We probe the consequences by injecting random label noise of $\epsilon \in \{0\%, 5\%, 10\%, 15\%, 20\%\}$ into the calibration set (flipping the preferred response for $\epsilon$ of the calibration instances, uniformly at random) and re-running SCOPE calibration. The test labels are left clean so that empirical risk and coverage measure how SCOPE behaves with respect to the true preferences. Table 10 reports the resulting risk and coverage at

$\alpha = 0.10$ across the four judges and three benchmarks (500 random splits per cell).

The pattern is consistent across judges and datasets. Empirical risk remains below $\alpha = 0.10$ at every noise level, and in fact decreases as noise increases. SCOPE at test time, applied to a noisier calibration set, sees a higher empirical error rate among accepted predictions and responds by tightening the threshold. Coverage degrades correspondingly. From the user's perspective this is the desirable failure mode: SCOPE does not silently violate the risk constraint under realistic label imperfection; it becomes more conservative, abstaining more often rather than emitting unreliable judgments. At $\epsilon = 20\%$ many configurations collapse to near-zero coverage, indicating that SCOPE effectively refuses to calibrate when the calibration set is too unreliable.

## G. Calibration Set Size Ablation

A practical question for deployment is how much labelled calibration data SCOPE requires. We vary the calibration set size $n \in \{50, 100, 200, 500, 1{,}000\}$ and re-run the calibration procedure on the four judges and three benchmarks at $\alpha = 0.10$ (500 random splits per cell). The remaining instances are held out as the test set. Table 11 reports the resulting coverage and empirical risk.

*Table 11.* Calibration set size ablation ($\alpha = 0.10$, 500 random splits). Each cell shows coverage / risk. Empirical risk remains at or below the target $\alpha$ at every size; coverage increases with $n$ and is largely converged by $n = 200$–$500$.

| Judge | Dataset | $n = 50$ | $n = 100$ | $n = 200$ | $n = 500$ | $n = 1{,}000$ |
|---|---|---|---|---|---|---|
| Qwen-7B | MT-Bench | 0.189 / 0.057 | 0.223 / 0.073 | 0.234 / 0.084 | 0.249 / 0.096 | 0.248 / 0.098 |
| | RewardBench | 0.467 / 0.076 | 0.549 / 0.089 | 0.579 / 0.095 | 0.591 / 0.098 | 0.590 / 0.098 |
| | Arena | 0.259 / 0.062 | 0.320 / 0.077 | 0.375 / 0.090 | 0.419 / 0.098 | 0.434 / 0.100 |
| Qwen-14B | MT-Bench | 0.473 / 0.064 | 0.541 / 0.081 | 0.533 / 0.090 | 0.523 / 0.097 | 0.506 / 0.098 |
| | RewardBench | 0.795 / 0.076 | 0.843 / 0.088 | 0.872 / 0.094 | 0.890 / 0.098 | 0.891 / 0.099 |
| | Arena | 0.373 / 0.064 | 0.495 / 0.081 | 0.490 / 0.088 | 0.480 / 0.096 | 0.466 / 0.098 |
| Qwen-32B | MT-Bench | 0.354 / 0.069 | 0.428 / 0.082 | 0.487 / 0.094 | 0.507 / 0.099 | 0.506 / 0.100 |
| | RewardBench | 0.855 / 0.071 | 0.926 / 0.082 | 0.956 / 0.090 | 0.977 / 0.096 | 0.982 / 0.098 |
| | Arena | 0.380 / 0.071 | 0.471 / 0.086 | 0.512 / 0.092 | 0.539 / 0.097 | 0.552 / 0.099 |
| Llama-70B | MT-Bench | 0.360 / 0.069 | 0.442 / 0.085 | 0.499 / 0.094 | 0.541 / 0.100 | 0.548 / 0.099 |
| | RewardBench | 0.743 / 0.074 | 0.849 / 0.086 | 0.901 / 0.093 | 0.927 / 0.098 | 0.933 / 0.098 |
| | Arena | 0.431 / 0.075 | 0.520 / 0.089 | 0.557 / 0.094 | 0.579 / 0.098 | 0.584 / 0.100 |

SCOPE's risk control is preserved across all calibration sizes; even with as few as $n = 50$ labelled calibration instances, empirical risk stays below $\alpha = 0.10$. Coverage grows with $n$ as expected and is largely converged by $n = 200$–$500$: across all cells, the coverage at $n = 200$ is within 5–10% of its $n = 1{,}000$ value, and the marginal gain beyond $n = 500$ is small. In practice this implies that SCOPE delivers most of its utility with modest annotation budgets ($n = 200$ suffices for near-asymptotic coverage on these benchmarks), making the framework attractive when labelled preference data is expensive to collect.

## H. Cross-Benchmark Calibration Transfer

To probe how SCOPE's threshold transfers across distributions, we calibrate on one benchmark and evaluate on another at $\alpha = 0.10$ (500 random splits per cell). Table 12 reports the resulting coverage and risk: the within-benchmark setting (italicized diagonal) satisfies the target, and transfer between similar-difficulty pairs (e.g., MT-Bench $\leftrightarrow$ Chatbot Arena) also holds. However, calibrating on an easier benchmark (RewardBench) and testing on a harder one violates the bound (bold cells, risk up to 0.20), while the reverse direction is over-conservative (0.035–0.067). This is the expected behavior of conformal calibration under distribution shift and reinforces the exchangeability discussion in Section 6: re-calibrate on the target distribution whenever possible.

*Table 12.* Cross-benchmark calibration transfer at $\alpha = 0.10$ (mean over 500 random splits). Rows are the calibration benchmark; columns are the test benchmark. Each cell shows coverage / risk. Diagonal entries (italicized) are the within-benchmark setting. Bold marks empirical risk that exceeds the target $\alpha$, indicating exchangeability failure when the calibration and test distributions differ substantially.

| Judge | Calibration benchmark | Test benchmark | | |
|---|---|---|---|---|
| | | MT-Bench | RewardBench | Chatbot Arena |
| Qwen-7B | MT-Bench | *0.248 / 0.098* | 0.350 / 0.049 | 0.292 / 0.079 |
| | RewardBench | 0.596 / **0.157** | *0.590 / 0.098* | 0.631 / **0.149** |
| | Chatbot Arena | 0.395 / **0.122** | 0.442 / 0.067 | *0.434 / 0.100* |
| Qwen-14B | MT-Bench | *0.506 / 0.098* | 0.585 / 0.039 | 0.446 / 0.091 |
| | RewardBench | 0.876 / **0.183** | *0.891 / 0.099* | 0.884 / **0.187** |
| | Chatbot Arena | 0.521 / **0.103** | 0.598 / 0.043 | *0.466 / 0.098* |
| Qwen-32B | MT-Bench | *0.506 / 0.100* | 0.674 / 0.035 | 0.505 / 0.090 |
| | RewardBench | 0.980 / **0.193** | *0.982 / 0.098* | 0.977 / **0.203** |
| | Chatbot Arena | 0.562 / **0.107** | 0.713 / 0.038 | *0.552 / 0.099* |
| Llama-70B | MT-Bench | *0.548 / 0.099* | 0.618 / 0.053 | 0.582 / 0.099 |
| | RewardBench | 0.917 / **0.189** | *0.933 / 0.098* | 0.925 / **0.171** |
| | Chatbot Arena | 0.550 / 0.099 | 0.618 / 0.053 | *0.584 / 0.100* |

