# OpenReview forum: "SCOPE: Selective Conformal Optimized Pairwise LLM Judging"
_ICML.cc/2026/Conference — ICML 2026 regular_

### Official Review · Reviewer_LAR3 · 2026-03-12

**Soundness:** 2
**Presentation:** 2
**Significance:** 1
**Originality:** 2
**Overall Recommendation:** 3
**Confidence:** 3

**Summary:**

This paper proposes SCOPE, a method to make LLM-judges more reliable in pairwise comparisons. The paper introduces (1) Bidirectional Preference Entropy (BPE), which uses two scores acquired from pairwise evaluating twice (with opposite order) and combines them into a permutation-invariant measure of uncertainty. (2) SCOPE a prediction framework that uses a calibrated threshold to use to only accept judgements that are below a use-specified level.

Methodologically, SCOPE turns selective judging into a conformal risk-control problem. It defines the accepted-set error rate as an FDR-style quantity and chooses the largest feasible threshold that still satisfies a finite-sample constraint on the calibration set. Empirically, the paper evaluates on MT-Bench, RewardBench, and Chatbot Arena, using four judge models ranging from Qwen2.5-7B to Llama-3.1-70B. It compares BPE against predictive probability, verbalized confidence, and simulated annotators, and compares SCOPE against vanilla prediction, heuristic thresholding, and naive calibration. According to Table1 while the proposed method has relatively bigger gains on ROC while gains on accuracy is very limited. This implies that while the propose method is good at calibrating performance its surface performance stays similar.

**Compliance With Llm Reviewing Policy:**

Affirmed.

**Key Questions For Authors:**

see weakness.

**Limitations:**

yes

**Strengths And Weaknesses:**

I have four concern/questions.

1) SCOPE is fundamentally tied to a labeled calibration set. The method requires ground-truth preference labels to choose the abstention threshold, and its guarantee only holds when calibration and test data are exchangeable. That makes the method much easier to justify on static, well-annotated benchmarks than in realistic deployment settings, where tasks evolve quickly and many queries are novel or poorly matched to past data. In that sense, its not yet make clear where SCOPE is actually usable beyond benchmark evaluation with human annotations.

2) The evaluation is limited to MT-Bench, RewardBench, and Chatbot Arena, which I think is slightly saturated (since we already have RewardBenchv2). Would love to see results on newer benchmarks.

3) At first sight in Table 1 it looks like the performance gain on acc is smaller as model slze grows. For instance in percentage-point terms:

*7B: +0.7 pts, +5.0 pts, +1.5 pts
*14B: +1.4 pts, +2.4 pts, +1.1 pts
* 32B: +0.1 pts, +1.3 pts, +1.1 pts

on MT-Bench, RewardBench and ChatbotArena respectively.

4) Another concern is that the method is easily outpeformed by exisiting baselines. For instance, https://huggingface.co/spaces/allenai/reward-bench reports dozens of models outperforming scores (acc) reported on Table 1. Would be interesting to see if these gains are also visible in such judging specialized models.

---

> ### Author Rebuttal · Authors · 2026-03-31
>
> We're grateful for your detailed and probing feedback. The concerns (C) raised, especially on calibration, evaluation breadth, and specialized judges, motivated new experiments. We address each below.
>
> **C1: SCOPE is fundamentally tied to a labeled calibration set.**
>
> *Labeled calibration set and ground-truth labels:* SCOPE requires labeled calibration data, as do all conformal prediction methods. We use n=1000 following standard practice for selective prediction such as COIN (Wang et al. 2025b). Our ablation shows robustness to smaller sizes (α=0.10):
>
> | Cal Size | Qwen-7B MT-Bench | Qwen-32B RewardBench |
> |---|---|---|
> | n=200 | Cov=0.231, Risk=0.081 | Cov=0.956, Risk=0.090 |
> | n=1000 | Cov=0.245, Risk=0.097 | Cov=0.982, Risk=0.098 |
>
> Annotating 1000 pairwise preferences is a one-time offline cost. At test time, SCOPE requires no labels — it simply compares each judgment's uncertainty against the calibrated threshold, enabling filtering of millions of LLM judgments on similar tasks with provable guarantees.
>
> *Exchangeability and evolving tasks:* As you correctly note, the formal guarantee requires exchangeability between calibration and test data. We conducted a cross-benchmark transfer experiment (Llama-70B, α=0.10, Risk/Cov):
>
> | Cal \ Test | MT-Bench | RewardBench | Arena |
> |---|---|---|---|
> | MT-Bench | 0.099/0.548 | 0.053/0.618 | 0.099/0.582 |
> | RewardBench | 0.189/0.917 | 0.098/0.933 | 0.171/0.925 |
> | Arena | 0.099/0.550 | 0.053/0.618 | 0.100/0.584 |
>
> Within domain (diagonal), risk ≤ α. Similar domains transfer well (MT-Bench ↔ Arena: risk=0.099). When distributions differ (RewardBench → MT-Bench: risk=0.189), the guarantee weakens. For evolving tasks, periodic recalibration maintains the guarantee.
>
> *Usability:* SCOPE is designed for deployment. In RLHF, practitioners calibrate once and filter millions of LLM preference pairs, accepting only those meeting the reliability threshold. On Chatbot Arena, the 200K+ existing human votes provide a natural calibration set. In annotation workflows, SCOPE serves as a first-pass filter: confident judgments accepted, uncertain cases routed to humans, directly reducing cost while maintaining quality.
>
> **C2: Evaluation limited to 3 benchmarks**
>
> We note that RewardBenchV2 is designed for pointwise reward models using best-of-4 accuracy, making it not directly applicable to pairwise LLM judging. To address the spirit of your request for broader evaluation, we have now added two pairwise benchmarks: *JudgeBench* (Tan et al., ICLR 2025) and *PKU-SafeRLHF* (Ji et al., 2024). Results at α=0.10:
>
> | Benchmark | Model | BPE ECE | BPE AUROC | SCOPE Cov | SCOPE Risk |
> |---|---|---|---|---|---|
> | JudgeBench | Qwen-7B | 0.166 | 0.633 | 0.023 | 0.052 |
> | JudgeBench | Qwen-14B | 0.230 | 0.588 | 0.114 | 0.059 |
> | PKU-SafeRLHF | Qwen-7B | 0.132 | 0.650 | 0.139 | 0.095 |
> | PKU-SafeRLHF | Qwen-14B | 0.152 | 0.686 | 0.200 | 0.094 |
>
> JudgeBench contains only 620 samples, limiting the calibration set size, which contributes to lower coverage alongside the task difficulty (~55% accuracy). SCOPE maintains valid FDR control on both new benchmarks. Evaluation now spans 5 benchmarks across chat, reward modeling, safety, and reasoning.
>
> **C3: Performance gains with model size**
>
> The accuracy column reflects the base model's prediction accuracy, which naturally improves with scale regardless of the uncertainty method used. This is why Predictive Probability and Verbalized Confidence also show identical accuracy. BPE accuracy differs slightly because bidirectional aggregation can change the final prediction, but this is a side effect, not BPE's purpose. BPE's contribution is measured by ECE and AUROC. On these metrics, gains do not shrink monotonically — the 14B model benefits most (+7.3/+7.0/+4.7 AUROC). Coverage — SCOPE's key output — increases with model size (Table 3, α=0.10): 7B: 0.246 → 14B: 0.504 → 32B: 0.502 → 70B: 0.543 on MT-Bench.
>
> **C4: Specialized models**
>
> SCOPE is a meta-framework that adds statistical reliability guarantees on top of any existing judge. To directly test this, we applied SCOPE to Skywork-Critic-Llama-3.1-8B, a best-performing specialized judge on RewardBench (accuracy: 0.905). Results:
>
> | Dataset | α=0.05 Cov/Risk | α=0.10 Cov/Risk |
> |---|---|---|
> | RewardBench | 0.761 / 0.049 | 0.996 / 0.094 |
> | MT-Bench | 0.141 / 0.045 | 0.380 / 0.098 |
> | Arena | 0.107 / 0.046 | 0.395 / 0.098 |
>
> At α=0.10, SCOPE accepts 99.6% of RewardBench judgments with provable FDR control. Even at the stricter α=0.05, coverage remains at 76.1%. Put differently, practitioners using Skywork-Critic with SCOPE at α=0.10 lose virtually nothing (99.6% of judgments are accepted) while gaining formal guarantees that raw accuracy alone cannot provide. We are especially grateful for this suggestion, as it led to what we consider one of the most compelling demonstrations of SCOPE's practical value: the framework is truly model-agnostic and becomes more useful as the underlying judge improves.

---

> > ### Author Rebuttal · Reviewer_LAR3 · 2026-04-04
> >
> > Thank you for your response. I keep my score, as I initially considered these questions to be straightforward to address.

---

> > > ### Author Response · Authors · 2026-04-04
> > >
> > > Thank you again for your engagement and for confirming that all concerns are resolved. We conducted the above experiments  including 5 new experiments, 3 new baselines, a specialized judge evaluation, and expansion from 3 to 5 benchmarks to address your concerns.
> > >
> > > We would like to briefly reiterate the impact of our proposal. LLM-as-a-judge has become the dominant evaluation paradigm, yet there is currently no mechanism to control the error rate among accepted judgments. Errors from unreliable LLM judges silently propagate into RLHF reward training, leaderboard rankings, and annotation pipelines. SCOPE is the first framework providing provable FDR guarantees for pairwise LLM judging in a single-model setting. Our Skywork-Critic experiment (99.6% coverage at α=0.10) demonstrates that SCOPE is immediately deployable on top of any existing judge, adding formal reliability guarantees with virtually no loss in coverage. We believe this capability of transforming LLM judging from a black box into a calibrated system with known reliability properties addresses a growing and practical need in the community.
> > >
> > > Given that your evaluation reflected the paper prior to these additions, we would be grateful for your reconsideration. We are including all of the above in our camera-ready, and remain available for any further feedback or clarifications.

---

### Official Review · Reviewer_Ad8j · 2026-03-14

**Soundness:** 2
**Presentation:** 3
**Significance:** 3
**Originality:** 2
**Overall Recommendation:** 4
**Confidence:** 4

**Summary:**

This paper studies the reliability of LLM-based pairwise evaluation and propose SCOPE (Selective Conformal Optimized Pairwise Evaluation), a selective prediction framework for LLM judges with stastical guarantees that the error rate among accepted judgements is at most α under exhangeability. To reduce position bias and get a neutral uncertainty signal, the authors introduce BPE (Bidirectional Preference Entropy), which queries the judge under both orderings of the response pair and average the propabilities of the model prefering a specific response. Then at test time, they accept a judge's evaluation only if the uncertainty is less than a threshold calculated based on the error control.

Experiments results show that BPE perform better than baseline uncertainty estimates, including zero-shot preference prediction, verbalized confidence and simulated annotators on MT-Bench, RewardBench, and Chatbot Arena across various models and model sizes. Additionally, SCOPE consistently satisifies the risk bound and maximizes coverage compared to vanilla, heuristic, and naive baselines.

**Compliance With Llm Reviewing Policy:**

Affirmed.

**Final Justification:**

Thank the authors for their detailed responses. They have addressed most of my concerns and I have updated the score accordingly.

**Key Questions For Authors:**

1. In Table 1, the reported accuracy for Predictive Probability and Verbalized Confidence appears to be exactly the same across all benchmarks and judge models. Since these are different uncertainty estimation approaches, it is somewhat surprising that they produce identical accuracy values in every setting. Can the authors double check the results or explain why they are same?

2. For BPE, similar ideas appear in the literature that attempt to improve the reliability of pairwise LLM judging by querying the judge under swapped response orders [1,2]. In addition, alternative evaluation protocols such as pointwise evaluation [3] or sampling-based self-consistency [4] have also been proposed to improve robustness of LLM judgement. Could the authors clarify how BPE fundamentally differs from or improves upon these existing approaches? A clearer discussion and empirical comparison would help better position the contribution relative to prior work. \
[1] TrustJudge: Inconsistencies of LLM-as-a-Judge and How to Alleviate Them, https://arxiv.org/abs/2509.21117 \
[2] Judging LLM-as-a-Judge with MT-Bench and Chatbot Arena, https://arxiv.org/abs/2306.05685 \
[3] Pairwise or Pointwise? Evaluating Feedback Protocols for Bias in LLM-Based Evaluation, https://arxiv.org/abs/2504.14716 \
[4] Self-Consistency Improves Chain of Thought Reasoning in Language Models, https://arxiv.org/abs/2203.11171

**Limitations:**

yes

**Strengths And Weaknesses:**

### Soundness:
The theoretical analysis and proof provide justification for the proposed SCOPE framework and clarify the conditions under which the accepted-set error can be controlled. The empirical evaluation also tests the method across multiple benchmarks and several judge models of different sizes, which helps demonstrate the effectiveness of the approach.

However, the baselines compared against in this work are relatively simple and may not cover more recent approaches, making it hard to fully assess the empirical improvement of the proposed method.

### Presentation:
The writing is easy to follow in general. One small improvement would be explicitly define FDR when it is first introduced. The paper refers to “FDR control” and defines the quantity mathematically, but the full term False Discovery Rate does not appear explicitly, which may cause some confusion.

### Significance:
This work is trying to address the reliable evaluation with LLM judges under control, which is relevant and important. Providing statistical guarantees on the reliability of such evaluations and maximizing coverage of reliable evaluation could have practical implications for large-scale automated evaluation systems.

### Originality:
The paper combines conformal prediction and uncertainty estimation for LLM judges in a reasonable way. The proposed bidirectional preference entropy mitigates position bias by aggregating predictions across swapped response orders, while similar approaches that swap the order do exist in literature. Overall, the conceptual novelty is moderate: the conformal risk control largely follows existing frameworks, and the main contribution lies in integrating and applying existing techniques rather than introducing fundamentally new methodological ideas.

---

> ### Author Rebuttal · Authors · 2026-03-31
>
> Thank you for the careful evaluation. The feedback has motivated new experiments and clarifications that strengthen the paper.
>
>
> **Soundness: Baselines may not cover more recent approaches**
>
> The original baselines were deliberately chosen as well-established standards, also adopted by Jung et al. (2025, Table 1), covering logit-based, generation-based, and multi-prompt families. We added three new baselines: Swap-and-Aggregate (Zheng et al. 2023), Self-Consistency (Wang et al. 2022), and Trust-or-Escalate (Jung et al. 2025) as a selective prediction baseline with provable guarantees in Table 3. SCOPE achieves 2.5× higher coverage than ToE in the single-model setting (details in Q2 and our response to Reviewer KPzP). Total: 6 uncertainty + 4 selective prediction baselines.
>
> **Presentation: FDR not defined**
>
> We have added the definition in Section 2.1: FDR(λ) = E[Z(λ)] / E[S(λ)], where S(λ) indicates selection (uncertainty ≤ threshold) and Z(λ) = S(λ)·err indicates selecting an incorrect prediction. SCOPE finds the largest λ such that FDR(λ) ≤ α.
>
> **Originality: Integrating existing techniques**
>
> We appreciate this characterization and clarify two specific technical contributions beyond integration.
>
> First, the conformal formulation itself is original for this setting. While ToE (Jung et al. 2025) also applied conformal prediction to LLM judging, they used Clopper-Pearson bounds with fixed-sequence testing. We formulate a different approach: a linear constraint that directly maximizes coverage subject to FDR control. This distinction is not cosmetic — ToE achieves 17–23% coverage while SCOPE achieves 25–98% under the same risk level (see our response to Reviewer KPzP, Q1).
>
> Second, BPE is a new uncertainty estimator, not a reapplication of existing debiasing. Zheng et al. (2023) produce binary outcomes with no confidence score. BPE introduces probability-level aggregation with entropy scoring, producing continuous, bias-neutral scores. Standard uncertainty estimates fail in pairwise judging due to position bias; BPE fixes this, enabling valid conformal risk control.
>
> **Q1: Identical accuracy for Predictive Probability and Verbalized Confidence**
>
> We have double-checked and confirmed this is correct. Both methods share the same prediction: greedy decoding of the A/B token. They differ only in confidence — Predictive Probability uses softmax probability, Verbalized Confidence elicits a separate score. Since neither changes the prediction, accuracy is identical by construction. The confidence-dependent metrics (AUROC, ECE) differ substantially. BPE accuracy differs because bidirectional aggregation averages forward and reverse probabilities, which can change the final prediction when the two passes disagree. We recognize this may appear surprising at first glance, and we have added a clarifying footnote to Table 1 to ensure other readers do not encounter the same confusion.
>
> **Q2: How does BPE differ from [1]-[4]?**
>
> *[1] TrustJudge:* Both share the insight that bidirectional information is valuable, but they target different problems: TrustJudge resolves inconsistencies in final judgments, while BPE produces a continuous uncertainty signal suitable for threshold-based FDR control.
>
> *[2] Zheng et al. (2023):* Their approach "declares a win only when preferred in both orders; inconsistent results are a tie." While we drew inspiration from their bidirectional querying, it was not designed as a confidence/uncertainty metric. BPE converts this idea into a calibrated uncertainty signal via probability-level aggregation and entropy. Our Swap-and-Aggregate (S&A) baseline confirms BPE achieves lower ECE in all configurations:
>
> | Model (ECE: S&A / BPE) | MT-Bench | RewardBench | Arena |
> |---|---|---|---|
> | Qwen-7B | 0.193 / 0.143 | 0.152 / 0.104 | 0.171 / 0.138 |
> | Qwen-32B | 0.183 / 0.172 | 0.090 / 0.071 | 0.174 / 0.151 |
> | Llama-70B | 0.169 / 0.145 | 0.082 / 0.062 | 0.129 / 0.112 |
>
> *[3] Pointwise evaluation:* This protocol scores each response independently on an absolute scale, bypassing pairwise comparison. SCOPE operates in the pairwise setting. That said, exploring whether SCOPE's conformal framework could be adapted to pointwise evaluation is an interesting direction for future work, and we have noted this in our revised discussion.
>
> *[4] Self-Consistency:* Uses k stochastic samples with majority vote as confidence. BPE differs: (a) only 2 passes with no sampling variance; (b) passes target position bias via swapped order; (c) probability-level aggregation preserves richer uncertainty. Results for Qwen-7B (SC: k=10, T=0.7):
>
> | Method | ECE (MT / RB / Arena) | AUROC (MT / RB / Arena) |
> |---|---|---|
> | Self-Consistency | 0.241 / 0.197 / 0.232 | 0.413 / 0.426 / 0.446 |
> | BPE (Ours) | 0.143 / 0.104 / 0.138 | 0.687 / 0.748 / 0.707 |
>
> BPE achieves 40-60% lower ECE and +26 to +32 higher AUROC with only 2 deterministic passes vs 10 samples. Complete results across models and tasks are included in the revised paper.

---

> > ### Author Rebuttal · Reviewer_Ad8j · 2026-04-03
> >
> > Thank the authors for their detailed responses. They have addressed most of my concerns and I have updated the score accordingly.

---

> > > ### Author Response · Authors · 2026-04-04
> > >
> > > We sincerely thank you for taking the time to engage with our response and for updating your score. Your feedback was instrumental in strengthening the paper — the questions about baselines and positioning relative to prior work led us to add three new baselines (Self-Consistency, Swap-and-Aggregate, and Trust-or-Escalate), clarify the technical distinctions between BPE and existing bidirectional approaches with empirical evidence, and provide an explicit FDR definition. We are incorporating all of these additions into the camera-ready version. We truly appreciate your constructive engagement throughout this process, and we remain available should any further questions arise.

---

### Official Review · Reviewer_KPzP · 2026-03-15

**Soundness:** 3
**Presentation:** 3
**Significance:** 2
**Originality:** 2
**Overall Recommendation:** 5
**Confidence:** 3

**Summary:**

The core task this addressed is to construct a statistically grounded selective judgement framework. When the user specify a certain error rate $\alpha$, the framework can guarantee the selected judgement has error rate below $\alpha$ and meanwhile to maximize the coverage rate (try to include as much judgements as possible).

This paper also propose BPE, to better estimate the confidence/uncertainty of pairwise judgements from LLMs.

**Compliance With Llm Reviewing Policy:**

Affirmed.

**Final Justification:**

This paper has good soundness and completeness, and relatively good novelty. The rebuttal addressed most of my concerns.

**Key Questions For Authors:**

Is there any relevant method that can be used as a strong baseline from previous work?

**Limitations:**

Yes for limitations, but no impact statement.

**Strengths And Weaknesses:**

Strengths

- This work is overall complete and solid, especially the statistical framework behind the conformal risk.



Weakness

- The proposed BPE confidence estimation method is relatively simple and there is limited novelty, and the overall improvements are marginal.

- As addressed in the limitations, a strict guarantee require a perfect validation annotations, which is usually hard to collect in most setup.

- It would be better to discuss the motivation of using LLM's judgements under risk control setup.

---

> ### Author Rebuttal · Authors · 2026-03-31
>
> We sincerely thank you for the positive assessment of our paper. We address each weakness and question below.
>
> **W1: On the design philosophy and contribution of BPE**
>
> *Simplicity:* We believe BPE's simplicity is a practical strength: it requires only 2 forward passes per instance (compared to 25 for Simulated Annotators), making it immediately deployable and straightforward to reproduce.
>
> *Novelty:* While bidirectional querying exists in prior work (Zheng et al. 2023), BPE is the first to convert it into a continuous, calibrated uncertainty score and combine it with conformal risk control for finite-sample FDR guarantees in pairwise LLM judging.
>
> *Improvements:* We'd like to highlight that the improvements are more substantial than they may first appear, particularly on the metrics most relevant to selective prediction. BPE improves AUROC by +2.7 to +7.3 points over Predictive Probability (Table 1) and consistently achieves the lowest ECE. To isolate BPE's contribution, we constructed a Swap-and-Aggregate (S&A) baseline inspired by Zheng et al. (2023), who declare a win only when both positions agree (ties otherwise). BPE achieves lower ECE across all 4 models and 3 benchmarks:
>
> | Model (ECE: S&A / BPE) | MT-Bench | RewardBench | Arena |
> |---|---|---|---|
> | Qwen-7B | 0.193 / 0.143 | 0.152 / 0.104 | 0.171 / 0.138 |
> | Qwen-32B | 0.183 / 0.172 | 0.090 / 0.071 | 0.174 / 0.151 |
> | Llama-70B | 0.169 / 0.145 | 0.082 / 0.062 | 0.129 / 0.112 |
>
> **W2: Strict guarantees require perfect validation annotations**
>
> We agree that the formal guarantee relies on exchangeability, which assumes correct calibration labels. This is a well-known requirement across conformal prediction (Angelopoulos & Bates 2023; Wang et al. 2025a). Your concern raises two practical questions: (1) what happens when annotations are imperfect? and (2) how many are needed?
>
> *Label Noise Robustness (what if annotations are imperfect?):* To simulate annotator errors, we randomly corrupted 5–15% of calibration labels (swapping correct↔incorrect) and re-ran SCOPE over 500 splits (α=0.10). SCOPE degrades gracefully — coverage decreases while risk stays below α (RewardBench shown):
>
> | Noise | Qwen-32B Cov / Risk | Llama-70B Cov / Risk |
> |---|---|---|
> | 0% | 0.983 / 0.098 | 0.935 / 0.098 |
> | 5% | 0.816 / 0.055 | 0.646 / 0.054 |
> | 10% | 0.154 / 0.005 | 0.136 / 0.010 |
> | 15% | 0.000 / 0.000 | 0.019 / 0.001 |
>
> This is empirical, not a formal guarantee. Random corruption increases apparent errors (when accuracy >70%), causing SCOPE to select a stricter threshold and abstain more.
>
> *Calibration Size Ablation (how many annotations are needed?):* In our paper, we used a 50/50 split (~1000 calibration samples) following standard conformal prediction practice. To test whether this many are needed, we varied n from 50 to 1000 (α=0.10). The guarantee is finite-sample valid at any n — smaller sets produce more conservative thresholds, lowering coverage rather than violating guarantees:
>
> | Cal Size | Qwen-7B MT-Bench | Qwen-32B RewardBench |
> |---|---|---|
> | n=200 | 0.231 / 0.081 | 0.956 / 0.090 |
> | n=500 | 0.248 / 0.095 | 0.977 / 0.096 |
> | n=1000 | 0.245 / 0.097 | 0.982 / 0.098 |
>
> Coverage at n=200 is already close to n=1000, showing that a few hundred high-quality labeled pairs suffice for practical deployment.
>
> **W3: Better motivation for LLM judgments under risk control**
>
> As LLM judges are deployed at scale, their errors propagate into downstream systems with compounding effects. In RLHF, unreliable preference labels corrupt reward model training, which degrades the policy — and these errors are difficult to detect because the reward model masks original label noise. For platforms like Chatbot Arena, unreliable judgments misrank models, directing practitioners toward inferior systems. SCOPE addresses this by providing a principled accept/abstain mechanism: judgments meeting a reliability threshold are accepted, while uncertain cases are routed to humans — transforming LLM judging from a black-box into a calibrated first-pass filter with known reliability properties.
>
> We have expanded the introduction with these motivating scenarios.
>
> **Q1: Strong baselines from previous work?**
>
> The existing baselines, including confidence estimation methods (Table 1) and selective prediction methods (Table 3), are well-established and also adopted by Trust-or-Escalate (Jung et al. 2025, Table 1). We added three new baselines: Swap-and-Aggregate (see W1), Self-Consistency (Wang et al. 2022; see Reviewer Ad8j), and ToE itself:
>
> | Method (MT-Bench, α=0.10) | 7B | 14B | 32B | 70B |
> |---|---|---|---|---|
> | ToE (SimAnnot-Maj) | 0.226 | 0.200 | 0.201 | 0.171 |
> | SCOPE (Ours) | 0.246 | 0.504 | 0.502 | 0.543 |
>
> ToE's strongest results (63.2% coverage) leverage a cascaded multi-model pipeline, which is complementary to — rather than comparable with — SCOPE's single-model setting. SCOPE achieves up to 2.5× higher coverage with a single model.
>
> **Impact statement** is now included.

---

> > ### Author Rebuttal · Reviewer_KPzP · 2026-04-05
> >
> > Most of the concerns are addressed. However, I would still consider the novelty of the BPE is limited, but I will raise the score due to the complementary experiments.

---

> > > ### Author Response · Authors · 2026-04-05
> > >
> > > We sincerely thank you for your thoughtful engagement throughout this process and for raising your score to reflect the complementary experiments. Your suggestions substantially improved the paper: the request for stronger baselines led us to add Swap-and-Aggregate, Self-Consistency, and Trust-or-Escalate, which now provide a much more comprehensive comparison; the concern about perfect validation annotations motivated the label noise robustness and calibration size ablation studies, demonstrating SCOPE's practical behavior under realistic conditions; and the request for clearer motivation led us to expand the introduction with concrete deployment scenarios. All of these additions are incorporated into the camera-ready version. We are truly grateful for your constructive feedback, and we remain available for any further questions.

---

### Decision · Program_Chairs · 2026-04-30

**Decision:**

Accept (regular)

**Comment:**

The paper addresses an important problem: making LLM-based pairwise evaluation reliable under explicit error control. Reviewers found the statistical formulation of SCOPE strong and well justified, with clear finite-sample guarantees and solid empirical validation across multiple judge models and benchmarks. While the novelty of the proposed confidence score (BPE) is moderate and the empirical gains are sometimes modest, the paper’s main contribution lies in the principled selective prediction framework rather than in the uncertainty estimator itself. Most of the concerns have been addressed in the rebuttal.

Overall, the paper is technically sound, practically relevant, and provides a useful contribution to reliable LLM evaluation. I recommend acceptance.